# Specific fibroblast subpopulations and neuronal structures provide local sources of Vegfc-processing components during zebrafish lymphangiogenesis

Guangxia Wang [1,2,3], Lars Muhl [4], Yvonne Padberg [1,2,3], Laura Dupont[5], Josi Peterson-Maduro[6], Martin Stehling[7], Ferdinand le Noble[8,9,10], Alain Colige[5], Christer Betsholtz[4,11], Stefan Schulte-Merker [1,2,3 ✉] & Andreas van Impel [1,2,3 ✉]

Proteolytical processing of the growth factor VEGFC through the concerted activity of CCBE1 and ADAMTS3 is required for lymphatic development to occur. How these factors act together in time and space, and which cell types produce these factors is not understood. Here we assess the function of Adamts3 and the related protease Adamts14 during zebrafish lymphangiogenesis and show both proteins to be able to process Vegfc. Only the simultaneous loss of both protein functions results in lymphatic defects identical to *vegfc* loss-of-function situations. Cell transplantation experiments demonstrate neuronal structures and/or fibroblasts to constitute cellular sources not only for both proteases but also for Ccbe1 and Vegfc. We further show that this locally restricted Vegfc maturation is needed to trigger normal lymphatic sprouting and directional migration. Our data provide a single-cell resolution model for establishing secretion and processing hubs for Vegfc during developmental lymphangiogenesis.

[1] Institute for Cardiovascular Organogenesis and Regeneration, WWU Münster, Münster, Germany. [2] Faculty of Medicine, WWU Münster, Münster, Germany. [3] Cells-in-Motion Cluster of Excellence, WWU Münster, Münster, Germany. [4] Integrated Cardio Metabolic Centre, Department of Medicine, Karolinska Institutet, Huddinge, Sweden. [5] Laboratory of Connective Tissue Biology, GIGA, University of Liège, Liege, Belgium. [6] Hubrecht Institute–KNAW & UMC Utrecht, Utrecht, The Netherlands. [7] Flow Cytometry Unit, Max Planck Institute for Molecular Biomedicine, Münster, Germany. [8] Department of Cell and Developmental Biology, Zoological Institute and Institute of Biological and Chemical Systems, Karlsruhe Institute of Technology (KIT), Karlsruhe, Germany. [9] Institute of Experimental Cardiology, University of Heidelberg, Heidelberg, Germany. [10] DZHK (German Center for Cardiovascular Research) partner site, Heidelberg/Mannheim, Germany. [11] Department of Immunology, Genetics and Pathology, Rudbeck Laboratory, Uppsala University, Uppsala, Sweden. ✉email: schultes@ukmuenster.de; vanimpel@uni-muenster.de

The lymphatic vasculature fulfills pivotal functions for maintaining tissue fluid homeostasis, for providing immune surveillance by transporting immune cells and soluble antigens, and for uptake and transport of dietary lipids from the small intestine. Lymphatic vessels form through a dynamic process called lymphangiogenesis which comprises sprouting and migration of lymphatic precursor cells from the venous endothelium leading to the formation of the first lymphatic structures that are connected to the venous system[1].

Zebrafish lymphatic development is tightly linked to the formation of intersegmental veins in the trunk. At about 32 hours post fertilization (hpf), sprouts emerge from the posterior cardinal vein (PCV). Almost half of these venous or secondary sprouts establish connections with arterial intersegmental vessels (aISVs) resulting in their remodeling into venous ISVs (vISVs). The other half of the sprouts migrate dorsally to the horizontal myoseptum (HM), where they transiently form a pool of lymphatic precursor cells named parachordal lymphangioblasts (PLs) at 48 hpf. Slightly later, at around 72 hpf, these PLs leave the HM region and migrate along aISVs in dorsal and ventral directions to form the trunk lymphatic vasculature, with the main lymphatic vessel (thoracic duct; TD) situated between the dorsal aorta (DA) and the PCV[2,3].

The key chemo-attractant for lymphatic endothelial cells (LECs) is vascular endothelial growth factor C (VEGFC). VEGFC is produced as a pre-pro-protein which, upon secretion, gets proteolytically cleaved at its C terminus by protein convertases such as Furin[4]. To obtain full biological activity, a second, N-terminal processing step is required before then the fully mature VEGFC binds with high affinity to vascular endothelial growth factor receptor 3 (VEGFR3 or Flt4), triggering downstream signaling events[5]. The extracellular VEGFC-processing machinery comprises collagen and calcium-binding EGF domain 1 (CCBE1), an essential component for processing of pro-VEGFC[6–10]. The actual processing step is mediated by A disintegrin and metalloprotease with thrombospondin motifs-3 (ADAMTS3)[8]. Loss-of-function of the gene leads to impaired lymphatic development in mice[11,12] and lymphedema formation in human patients[13].

The genetic conservation of VEGFC signaling components is remarkable, with mutations in the respective zebrafish and murine genes leading to a complete absence of lymphatic structures[14,15]. Furthermore, mutations in the human orthologues of these genes have been identified as causative for several congenital lymphedema syndromes[16]. The cellular sources of these proteins, however, remain enigmatic, and given the importance of these secreted factors for lymphangiogenesis, it is surprising that neither the tissue distribution of VEGFC, CCBE1, and ADAMTS3, nor their exact interaction in time and space has been resolved. Here we address the question how Vegfc, Ccbe1, Adamts3, and Adamts14 coordinate spatially and temporally to regulate lymphangiogenesis in the zebrafish embryo. We find that *adamts3* and *adamts14* are both sufficient to activate Vegfc in vivo, and only double mutants phenocopy the *vegfc* mutant phenotype. Expression analysis of these proteases identifies that both enzymes are not only provided by neuronal tissues but furthermore by cells close to the migration path taken by the secondary sprouts, including expression in the HM region. Using cell transplantation approaches, we find that both neuronal as well as non-neuronal expression domains of either protease are sufficient to locally rescue sprouting defects in double mutants. In depth analysis of non-neuronal expression domains via single cell sequencing and analysis of several novel transgenic reporter lines revealed that a subpopulation of *pdgfra*-positive fibroblasts locally secretes both proteases at the HM. Importantly, the identical population of fibroblasts also provides Ccbe1 and Vegfc at the HM, allowing Vegfc release and activation in the region. We

furthermore show that the activation of Vegfc in this area is key for the migration of PL cells to the HM, suggesting a model in which the overlapping expression of *adamts3*, *adamts14*, *vegfc*, and *ccbe1* in the fibroblast population at the HM leads to local processing of Vegfc which in turn guides lymphatic precursor cells to this position within the embryo.

## Results

**Procollagen N-proteinase function during lymphangiogenesis.** In order to study Adamts3 function in zebrafish, we generated two *adamts3* knockout alleles harboring out-of-frame deletions in exon 3 (encoding the prodomain) or exon 5 (encoding the catalytic metalloproteinase domain) (Fig. 1a). For both alleles, crosses of heterozygous carriers did not yield embryos with lymphatic defects. At 48 hpf *adamts3* mutants showed normal PL formation (Fig. 1b–d), and a wild-type TD at 5 dpf (Fig. 1f–h). Given the strong evolutionary conservation of key pathway members of the Vegfc/Flt4 signaling axis, we wondered whether another related protease might be involved in Vegfc processing. Besides the two other members of the so-called procollagen N-proteinase family, *adamts2* and *adamts14*[17], an additional related gene exists in zebrafish, termed *adamts2_like*. We introduced frame-shift causing mutations in exons encoding the conserved catalytic metalloproteinase domains for all three genes (Fig. 1a, Supplementary Fig. 1a). Single mutants for any of these three genes did not interfere with lymphatic development (Fig. 1e, i, Supplementary Fig. 1b, c). However, analysis of all possible double mutant situations revealed lymphatic defects upon simultaneous loss of *adamts3* and *adamts14*, but in no other allelic combination (Fig. 1j). *adamts3;adamts14* double mutant embryos completely lacked PLs and, reminiscent of *vegfc* and *ccbe1* mutants, also lacked vISVs at 48 hpf, suggesting that venous sprouting is likewise completely blocked (Fig. 1k, l; Supplementary Movies 1 and 2). Double mutants also lacked all lymphatic structures in the trunk at 5 dpf (Fig. 1m). In embryos harboring at least one functional copy of either *adamts3* or *adamts14*, lymphatic development was unaffected (Supplementary Fig. 1d, e), suggesting that both proteins have redundant functions during secondary sprouting. Furthermore, we found that both facial lymphatic vessels[18,19] and also the recently discovered lymphatic cell population within the meningeal layer of the zebrafish brain[20–22] are completely absent in double mutant embryos (Fig. 1n, o). In addition, double mutants displayed a variable strong curvature of the trunk, which we occasionally also observed in *adamts3*$^{+/-}$*;adamts14*$^{-/-}$ embryos with wild-type lymphatics (Fig. 1p, q). We conclude that the activity of Adamts3 and Adamts14 is an essential component of Flt4-driven lymphatic development in zebrafish, with both genes acting redundantly.

**Adamts3 and Adamts14 are both able to process Pro-Vegfc.** The combined loss of Adamts3 and Adamts14 resembles defects seen when blocking the Vegfc signaling pathway, suggesting that both proteases mediate the processing of Pro-Vegfc during zebrafish lymphangiogenesis. In order to substantiate this notion, we took advantage of an in vivo Vegfc activity assay[23]. Here, the unprocessed, full-length form of zebrafish Vegfc is expressed ectopically from the floorplate, resulting in a strong venous hyper-branching phenotype in wild-type embryos (Fig. 2a, b). In mutants with reduced Vegfc processing, this dominant phenotype is suppressed[10]. Overexpression of Vegfc from the floorplate did not result in hyper-branching in *adamts3;adamts14* double mutants (Fig. 2c) which is in line with the complete loss of Adamts3 and Adamts14 activity resulting in a blockage of Vegfc processing. However, one functional copy of *adamts3* or *adamts14* seems sufficient to trigger the hyper-branching

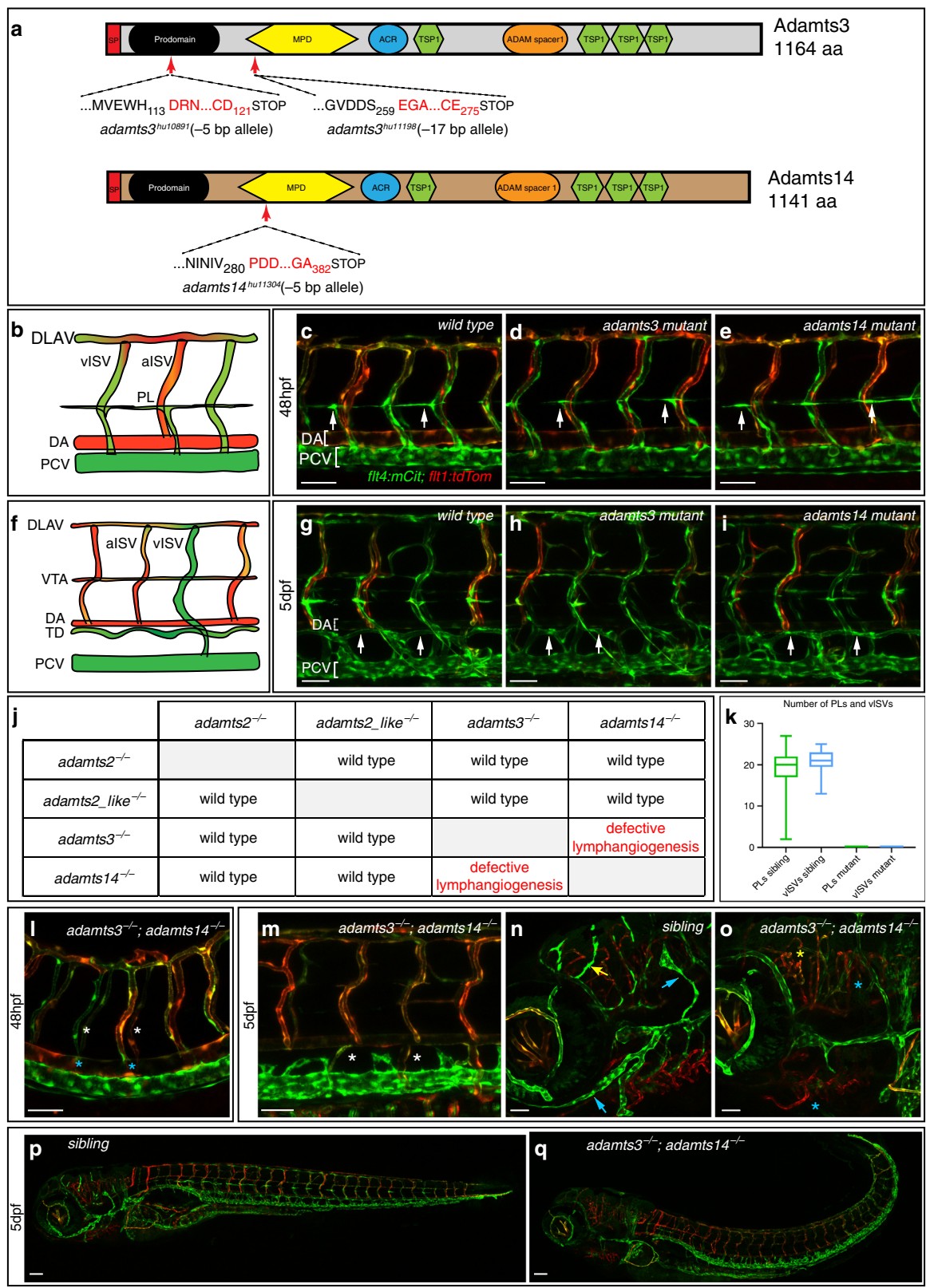

phenotype (Fig. 2d, e), demonstrating that both zebrafish proteins are able to process Vegfc.

Since previous reports suggested that human ADAMTS14 is unable to cleave VEGFC[8], we wondered whether the zebrafish enzyme is able to process human VEGFC in vivo. Transient floorplate expression of full-length human VEGFC resulted in venous hyper-branching in the proximity (Fig. 2f), as shown

previously[23]. Analogous to what we have seen for zebrafish Vegfc, both *adamts3* and *adamts14* single mutants still displayed dominant hyper-branching phenotypes, while the only allelic combination in which the effects were suppressed was *adamts3; adamts14* double mutant (Fig. 2g–i). Hence, zebrafish Adamts14 can cleave human VEGFC in vivo, which prompted us to re-assess the ability of human ADAMTS14 to process VEGFC in vitro.

**Fig. 1 Lymphangiogenesis is completely abolished in *adamts3; adamts14* double mutant embryos. a** Schematic of Adamts3 and Adamts14 protein structures depicting the predicted effects of the indicated deletion alleles. Shown are the aa positions of the deletion-induced frame shifts (red letters) as well as the position of the resulting premature stop codons. SP signal peptide, MPD metallopeptidase domain, ACR ADAM cysteine-rich domain 2, TSP1 thrombospondin type-1. **b** Schematic representation of the wild-type trunk vasculature at 48 hpf. **c–e**, **g–i**, and **l–q** *flt4:mCitrine; flt1:tdTomato* double transgenic embryos highlighting arterial ECs in red and venous and lymphatic structures in green. In wild-type (**c**), *adamts3* (**d**) or *adamts14* single mutants (**e**), PL cells align at the HM at 48 hpf (arrows). **f** Schematic representation of the trunk vasculature at 5 dpf with the thoracic duct (TD) being located between DA and PCV. Compared with wild type (**g**), neither homozygous *adamts3* (**h**) nor *adamts14* mutants (**i**) exhibit lymphatic defects at 5 dpf (arrows). **j** Table summarizing the analysis of lymphatic phenotypes in all double mutant combinations for *adamts2, adamts2_like, adamts3*, and *adamts14*. **k** At 48 hpf, the number of PLs was quantified in $n = 53$ siblings and $n = 30$ *adamts3; adamts14* double mutants and the number of vISVs was quantified in $n = 37$ siblings and $n = 11$ double mutants. Box-and-whisker plots show median and quartiles with whiskers indicating minimum/maximum values (source data are provided as a Source Data file). **l** *adamts3; adamts14* double mutants lack PLs (white asterisks) and do not form vISVs (blue asterisks) at 48 hpf. **m** At 5 dpf, the TD is completely absent in double mutants (white asterisks). Facial lymphatics (blue arrows) and meningeal lymphatics (yellow arrow) are formed in siblings at 5 dpf (**n**) but are absent in *adamts3; adamts14* double mutants (**o**). In contrast to their siblings (**p**), double mutants (**q**) display a variably strong curvature of the trunk. Scale bars in **c–e**, **g–i**, **l–o**: 50 μm; **p**, **q**: 100 μm. aa amino acid, bp base pair, dpf days post fertilization, ECs endothelial cells, DA dorsal aorta, DLAV dorsal longitudinal anastomotic vessel, hpf hours post fertilization, HM horizontal myoseptum, a/vISV arterial/venous intersegmental vessel, PCV posterior cardinal vein, PL parachordal lymphangioblast, VTA vertebral artery.

To this end, we used conditioned medium of HEK293 cells, expressing human pro-VEGFC in the presence of the serine protease inhibitor AEBSF (to prevent cutting by plasmin) and the furin inhibitor Decanoyl-RVKR-CMK (to retain significant amounts of unprocessed full-length VEGFC). Under these conditions, a full-length VEGFC band at 58 kDa and the C-terminally processed 31 kDa form can be detected, indicating that Furin inhibition is only partial (Fig. 2j). Addition of enriched recombinant ADAMTS3, however, resulted in a decrease of these 58 kDa and 31 kDa polypeptides and the emergence of a 45 kDa as well as of the fully mature 21 kDa form of VEGFC (Fig. 2j), consistent with the previously reported VEGFC cleavage activity of ADAMTS3[8,11]. Of note, N-terminal processing was also evident after incubation of the conditioned medium with enriched ADAMTS14 protein, indicating that also human ADAMTS14 has the capacity to cleave VEGFC. ADAMTS3- and ADAMTS14-dependent VEGFC processing was completely abolished by EDTA, an inhibitor of metalloproteinase activity. Taken together, this in vitro assay demonstrates that N-terminal processing of human VEGFC can not only be mediated by ADAMTS3 but also by ADAMTS14, a finding that is in line with human VEGFC overexpression phenotypes depending on zebrafish Adamts14 activity in vivo (Fig. 2i).

***adamts3* is expressed along the migration route of PL cells**. To examine the expression pattern of *adamts3*, we performed in situ hybridization (ISH) on embryos at 36 and 48 hpf (Fig. 3a–c). Consistent with previously published data[24], strong expression in the spinal cord was observed (Fig. 3a). We furthermore noticed weak expression in cells located around the main axial blood vessels, the DA and PCV, at both stages (Fig. 3b, c). To analyze *adamts3* expression in more detail, we generated an *adamts3: Gal4FF* reporter line. At 48 hpf, transgenic embryos showed prominent expression in the spinal cord and axonal extensions (Fig. 3d) of motoneurons co-expressing the motoneuronal marker *mnx1* (Fig. 3e-g). In zebrafish embryos, three different types of motoneurons exist per segment, namely the rostral primary motoneuron (RoP), the middle primary motoneuron (MiP), and the caudal primary motoneuron (CaP), all projecting into different parts of the musculature[25] (Fig. 3h). By using *kdrl:mCherry* to highlight endothelial cells and *adamts3:Gal4FF;UAS:GFP*, we observed that axonal extensions of *adamts3*-positive RoP motoneurons were positioned in close proximity to PL cells at the HM at 48 hpf, while *adamts3*-positive CaP axons extended further ventrally passing by the DA and PCV (Fig. 3i, m). Weaker *adamts3* expression domains were found within cells located at the segment boundaries along the migration route of lymphatic

precursor cells (Fig. 3i, j). We furthermore noticed *adamts3*-expressing cells lining the HM and ending up juxtaposed to PL cells (Fig. 3k, m, Supplementary Movie 3), as well as single *adamts3*-positive non-endothelial cells around the DA and PCV (Fig. 3l). Several of these expression domains (Fig. 3n), including the expression within RoP axons, correlate spatially with the migration routes of secondary sprouts from the PCV giving rise to both, vISVs and lymphatic precursors at the HM.

***adamts14* is expressed in the floorplate and at the HM**. We next analyzed *adamts14* expression patterns. At 26 hpf, shortly before the onset of venous sprouting, expression was most prominent in the floorplate as detected by ISH (Fig. 4a, b). In addition, we identified staining around the DA and PCV, along ventral segment boundaries (Fig. 4b), and at the HM level (Fig. 4c, d). During secondary sprouting stages at 36 hpf, *adamts14* expression remained strong in the floorplate and was detectable around the DA and PCV (Fig. 4e, f) and in cells located at the HM (Fig. 4g, h). At 48 hpf, when PLs have already reached the HM region, expression was still evident in the floorplate (Fig. 4i, j), in cells around the DA and PCV (Fig. 4j) and to a lesser extent at the HM (Fig. 4k, l). Since there are no *adamts14* BACs available covering the relevant genomic region, we decided to additionally perform RNAscope[26], and were indeed able to detect *adamts14* RNA granules at the level of the HM (Fig. 4m, n). In sum, *adamts14* is expressed by the floorplate, by tissue at the level of the main axial vessels and furthermore by cells at the HM during venous sprouting stages (Fig. 4o).

**Analysis of lymphatic-relevant *adamts3* expression domains**. To identify *adamts3* expression domains essential for venous sprouting, we aimed at generating tissue-specific *adamts3* knock-outs using CRISPR/Cas9-based approaches[27–29]. However, due to mosaic expression using this Gal4/UAS-driven technique, we were unable to efficiently target *adamts3* and did not observe a phenotype even when driving expression of the constructs ubiquitously in *adamts3*$^{+/-}$;*adamts14*$^{-/-}$ embryos. Since Adamts3 is a secreted enzyme, a mosaic deletion is presumably insufficient, as even a few *adamts3* wild-type cells per segment will likely provide relevant amounts of the protease.

We therefore continued by assessing which *adamts3* expression domains are sufficient to trigger venous sprouting and PL migration, and carried out a series of heterochronic cell transplantation experiments, transferring cells from donors mutant for *adamts14* (but wild-type for *adamts3*) into double mutant embryos. We used *adamts3:Gal4FF;UAS:GFP* transgenic

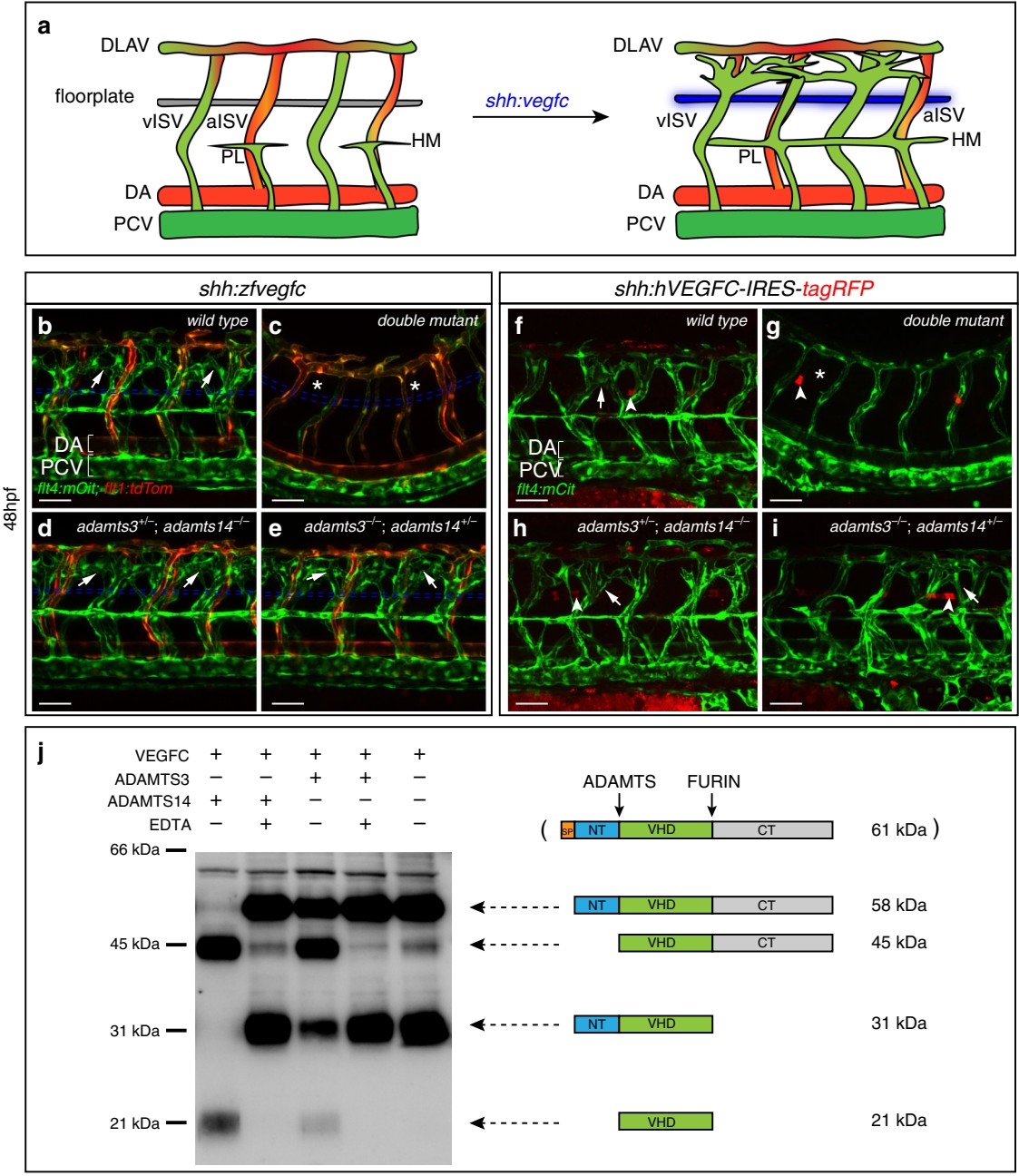

**Fig. 2 Adamts3 and Adamts14 are both able to process pro-Vegfc. a** Schematic depiction of the in vivo Vegfc-processing assay. In wild-type embryos, floorplate expression of full-length Vegfc results in venous hyper-branching at 48 hpf. **b–e** *flt4:mCitrine; flt1:tdTomato* transgenic embryos with arterial structures in red and venous and lymphatic structures in green. **b, c** While forced expression of full-length zebrafish Vegfc from the floorplate leads to hyper-branching in wild-type embryos (**b**, arrows), this dominant phenotype is not visible in *adamts3; adamts14* double mutants (**c**, asterisks). Allelic combinations with one remaining wild-type copy of either *adamts3* (**d**) or *adamts14* (**e**) develop the Vegfc-induced hyper-branching phenotype. Blue lines indicate the floorplate position (n = 225 analyzed embryos). **f–i** Mosaic expression of full-length human VEGFC in floorplate cells (marked by tagRFP expression) results in local venous hyper-branching in wild-type embryos (**f**), but not in *adamts3;adamts14* double mutants (**g**). One functional copy of either gene (**h, i**) is sufficient to restore the dominant phenotypes (n = 157 injected tagRFP+ embryos). Scale bars: 50 μm. DA dorsal aorta, DLAV dorsal longitudinal anastomotic vessel, hpf hours post fertilization, HM horizontal myoseptum, a/vISV arterial/venous intersegmental vessel, PCV posterior cardinal vein, PL parachordal lymphangioblast. **j** N-terminal processing of human VEGFC by ADAMTS3 and ADAMTS14 proteases. Conditioned medium from HEK293 cells expressing full-length VEGFC was incubated with ADAMTS3, ADAMTS14, or control buffer, in the presence or absence of EDTA. VEGFC forms were analyzed by Western blotting. In absence of active ADAMTS proteins (lanes 2, 4, and 5), VEGFC can be detected as a 58 kDa (full-length VEGFC without SP) and a 31 kDa form (C-terminally processed VEGFC without SP). In the presence of active ADAMTS14 (lane 1), the 58 kDa form is converted into a 45 kDa polypeptide while the 31 kDa form is processed into the fully mature 21 kDa VEGFC, which is in line with N-terminal processing of VEGFC proteins. A similar shift in VEGFC forms is detectable in the presence of active ADAMTS3 (lane 3) [source data are provided as a Source Data file]. Schematic depicting different VEGFC forms and their molecular weights. SP signal peptide, NT N-terminal propeptide, VHD VEGF homology domain, CT C-terminal propeptide.

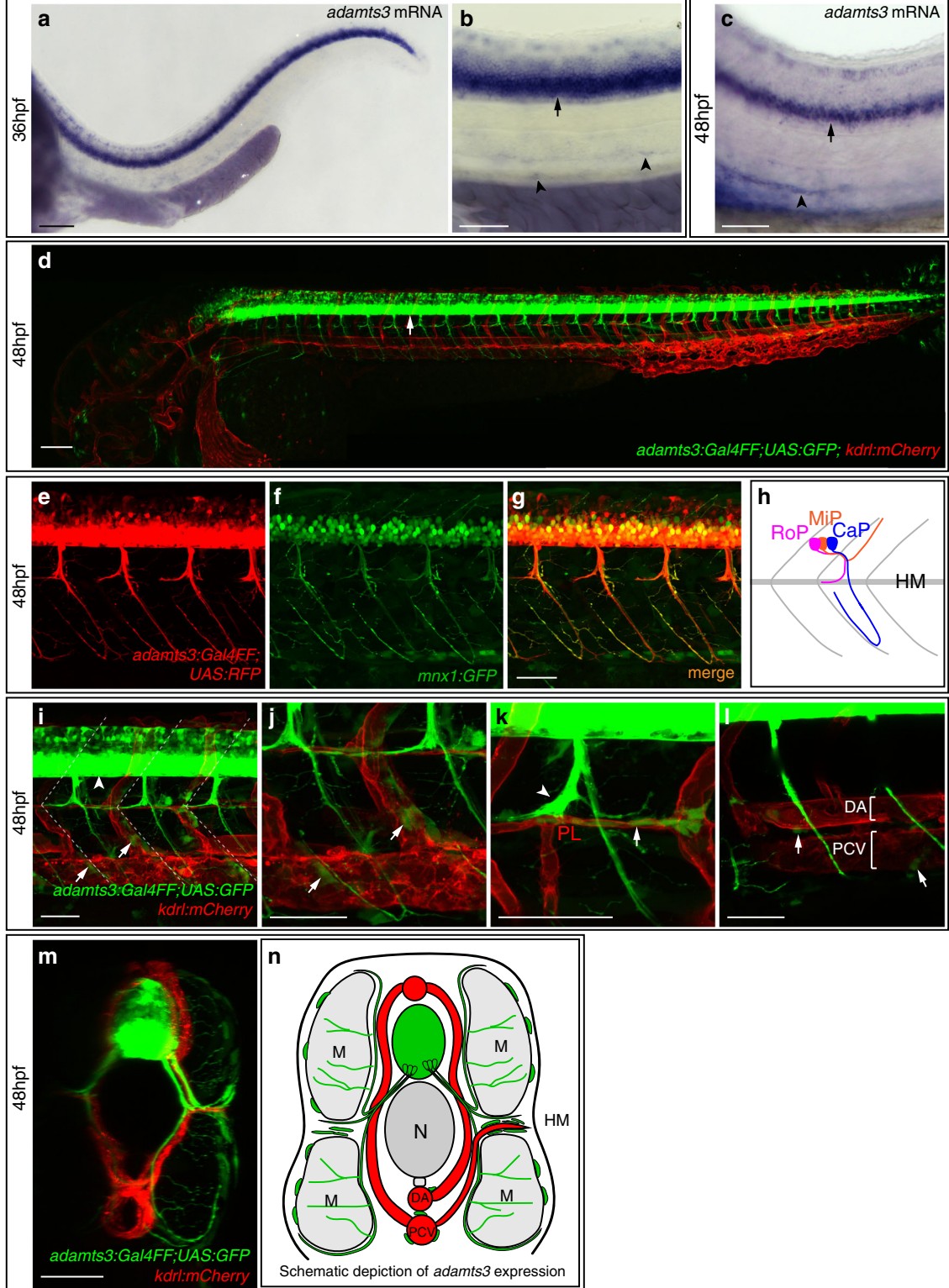

donors injected with Dextran-Alexa647 in order to trace all transplanted cells, while specifically highlighting those with *adamts3* expression (Fig. 5a). Since *adamts3;adamts14* double mutants do not exhibit any vISVs or PL cells (Fig. 1k), formation of those structures in double mutant host embryos would indicate rescuing capacity for the transplanted cells.

Previous work postulated a role for motoneurons in lymphatic guidance, since ablation of motoneurons was shown to cause defects in PL formation[30]. The molecules mediating this interaction, however, remained unknown. As *adamts3;adamts14* double mutants neither displayed motoneuron loss nor any axonal pathfinding defects (Supplementary Fig. 2), we wondered whether *adamts3* expression in motoneurons could affect sprouting and migration of venous ECs. In cell transplantations with *adamts3*+ RoP and CaP motoneurons, both PLs and vISVs were formed in the proximity of these neurons at 48 hpf (Fig. 5b, c, Supplementary Movie 4). This demonstrates that motoneuronal expression of *adamts3* is sufficient to trigger Vegfc processing in a fashion that

**Fig. 3 *adamts3* is expressed by motoneurons and additional cell types during venous sprouting stages. a–c** Whole mount in situ hybridization against *adamts3* at 36 and 48 hpf indicates strong expression in the spinal cord (arrow) and weaker expression of *adamts3* transcripts around the axial blood vessels in the trunk (arrowheads). **d** The *adamts3:Gal4FF; UAS:GFP* transgenic reporter line shows very prominent expression (in green) in neuronal cells (arrow). **e–g** The *adamts3:Gal4FF; UAS:RFP* transgene shows expression in trunk neurons and axons that are also labeled by the motoneuronal marker *mnx1: GFP* at 48 hpf. **h** Diagram of the three different types of motoneurons per segment, differing in their axonal projections which either extend along the HM region (rostral primary motoneuron, RoP, shown in magenta), toward the dorsal side of the trunk (middle primary motoneuron, MiP, highlighted in orange) or that extend further ventrally, passing by the main axial blood vessels (caudal primary motoneuron, CaP shown in blue). **i–l** *adamts3:Gal4FF; UAS:GFP; kdrl: mCherry* transgenic embryos, highlighting *adamts3* expression domains in green and ECs in red at 48 hpf. **i** Higher magnification of the trunk region, showing *adamts3* reporter expression not only in neurons (arrowhead) but also in non-neuronal cells located at the segment boundaries (arrows). White lines indicate the position of the segment boundaries. **j** Higher magnification of the ventral trunk region, indicating the position of *adamts3*-expressing cells along the segment boundaries (arrows). **k** Partial z-projection of the HM region indicating expression of the *adamts3* reporter (in green) in the RoP axon (arrowhead) and in cells along the HM (arrow), which are in close proximity to the PL cells (in red). **l** In partial z-projections of the trunk, individual *adamts3*-positive cells around the DA and the PCV are apparent (arrows). **m** Virtual cross section of the trunk region shown in **i**. **n** Schematic cross view illustrating the different *adamts3* expression domains (green) in the trunk.; Scale bar in **a**, **d**: 100 μm; **b**, **c**, **e–g**, **i–m**: 50 μm. DA dorsal aorta, ECs endothelial cells, hpf hours post fertilization, HM horizontal myoseptum, M muscle, N notochord, PCV posterior cardinal vein, PL parachordal lymphangioblast.

allows local rescue of the venous sprouting defects in double mutant embryos, establishing a direct molecular connection between embryonic motoneurons and the control of venous sprouting.

Independently, we also observed PL and intersegmental vein development in embryos in which transplants only contained non-neuronal *adamts3*+ cells populating the HM region, the segment boundary, and the DA and PCV region (Fig. 5d, e). In these instances, PLs were always juxtaposed to *adamts3*+ cells at the HM, suggesting a local relationship between the source of Adamts3 and the final PL position. When transplanted *adamts3*+ cells populated the segment boundary together with *adamts3*+ cells around the DA and PCV, both lymphatic precursor cells and vISVs were detectable in the immediate vicinity at 48 hpf (Fig. 5f, g, Supplementary Movie 5). Finally, in cases where transplants gave rise to *adamts3*-expressing cells both around the PCV and at the HM, a full local rescue with vISVs and PLs juxtaposed to the *adamts3*-positive cells at the HM was evident (Fig. 5h, i). Together, these findings demonstrate additional non-neuronal *adamts3* sources to be sufficient to locally induce sprouting from the PCV, leading to the formation of both veins and PL cells (Fig. 5j).

**Identification of essential expression domains of *adamts14*.** Analogous to what we did for *adamts3*, we performed transplantation experiments where *adamts3* mutant cells containing at least one *adamts14* wild-type copy were transplanted into *adamts3;adamts14* double mutant recipients (Fig. 6a). In the absence of a transgenic reporter construct, we relied on labeling donor cells by Dextran-Alexa647 injection. Transplants contributing to bigger patches of floorplate cells, which is the most prominent *adamts14* expression domain in the embryo, resulted in vISV formation and smaller endothelial sprouts at the HM in double mutants, reminiscent of normal PL cells (Fig. 6b–d, Supplementary Movie 6). Hence, Adamts14 provided from the floorplate can trigger sprouting of venous ECs from the PCV, identifying Adamts14 as a floorplate-derived factor influencing lymphangiogenesis.

Alternatively, when transplanted cells contributed to the myotome, we found two different scenarios: rescue of vISVs and PL formation was only evident in embryos in which transplants not only gave rise to muscle cells, but also to additional groups of Alexa647-positive cells located at the HM, next to a segment boundary and around the DA and PCV (Fig. 6e–h, Supplementary Movie 7), all regions in which we had detected *adamts14* expression before. Cases, however, in which transplants contributed exclusively to muscle, never showed rescue (Fig. 6i–k), suggesting that these additional non-floorplate cells of unknown identity represent the essential Adamts14 source

to provide rescuing activity. Again we noticed that PL cells appeared juxtaposed to transplanted cells at the HM region, similar to what we had observed for *adamts3* transplants (Fig. 6l–n, Supplementary Movie 8). In conclusion, besides the *adamts14* floorplate expression domain, also cells positioned around the migration route of venous sprouts close to the axial vessels, along segment boundaries, and at the HM are sufficient to provide Adamts14 to an extent that restores secondary sprout migration in a locally restricted fashion in double mutant embryos (Fig. 6o).

***adamts3*+ cells co-express *vegfc*, *ccbe1*, and *adamts14*.** The cell transplantation data suggested an involvement of thus far uncharacterized cell populations located at the HM, the segment boundaries and around the DA and PCV in terms of Vegfc-processing and enabling venous sprouting. To better understand which cell types express *adamts3*, we performed fluorescence-activated cell sorting (FACS) followed by RNA sequencing, using *adamts3:Gal4FF;UAS:GFP* transgenic embryos at 48 hpf. In order to enrich for non-neuronal *adamts3*+ cells we also sorted for the pan-neuronal transgene *Tg(NBT:dsRed)* reasoning that neuronal *adamts3*-expressing cells would be double positive while non-neuronal *adamts3*-expressing cell types should only exhibit a GFP signal (Supplementary Fig. 3a). Double negative cells served as a control population (Fig. 7a).

As expected from our ISH results, GFP+ dsRed+ neurons showed significantly higher *adamts3* levels than the non-fluorescent control population (Fig. 7b). The same cell population also showed twofold enrichment of *adamts14* expression compared with the negative population, possibly reflecting *adamts14* floorplate expression or the existence of a potential neuronal subpopulation expressing *adamts14*. When comparing the GFP+dsRed− population (non-neuronal *adamts3*-expressing cells) to the control, we detected the predicted enrichment for *adamts3* transcripts. This enrichment appeared lower compared with the neuronal population, which is consistent with the overall higher expression levels of *adamts3* in motoneurons (Fig. 3). *adamts3*+ cells also showed increased *adamts14* mRNA levels, suggesting that both genes are co-transcribed in non-neuronal cells, a notion in line with the transplantation results. We reasoned that other genes involved in the Vegfc-processing machinery might also be co-expressed in these cells and analyzed *ccbe1* expression levels. Previously we have reported that *ccbe1* is expressed in a dynamic fashion along the migration route of lymphatic precursor cells including expression along the segment boundaries and at the HM by mesodermal cells[6]. While co-expression of *adamts3* and *ccbe1* in GFP+dsRed+ neurons is not evident, we found a 3.3-fold *ccbe1* enrichment within the

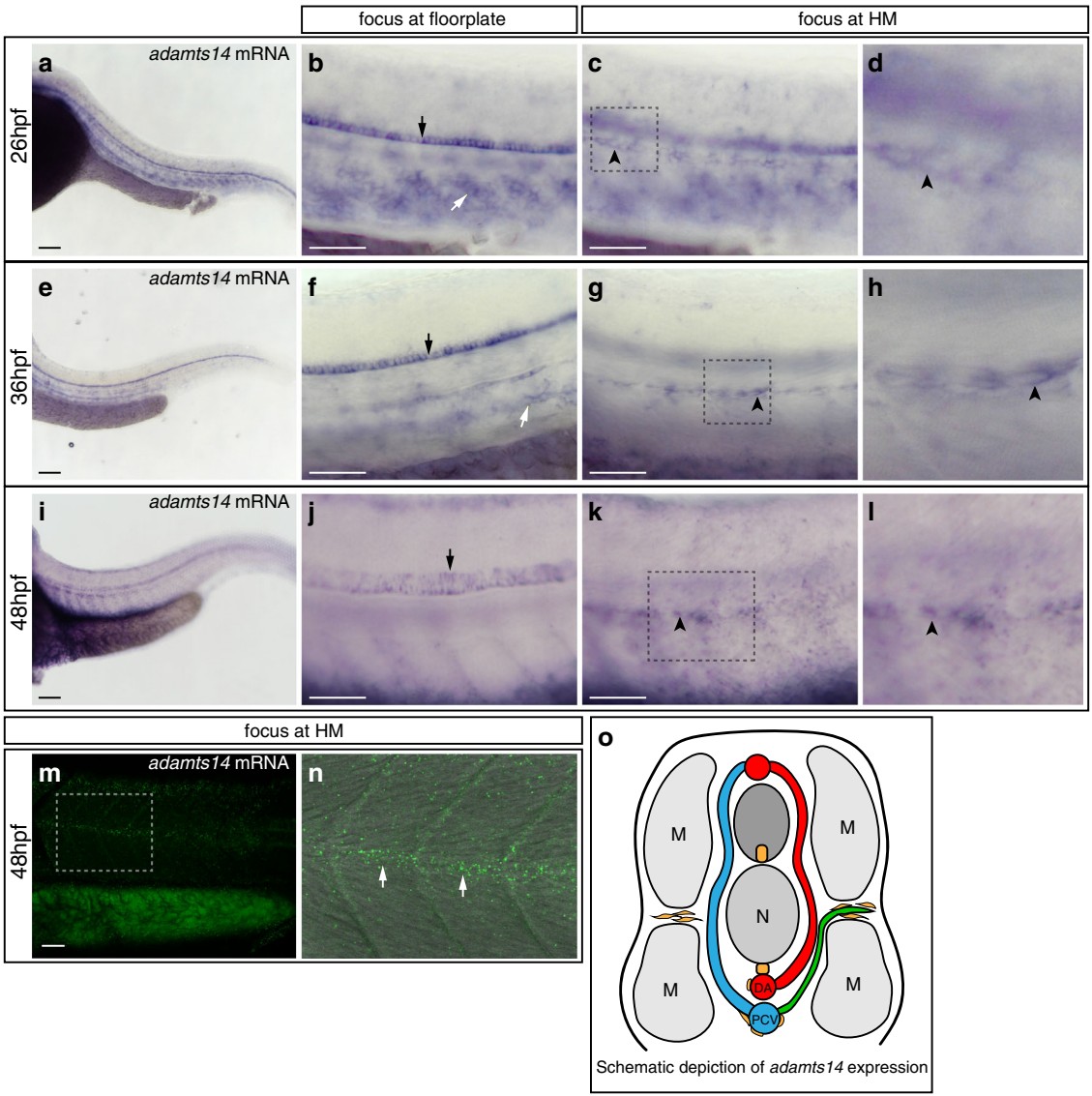

**Fig. 4 *adamts14* expression can be detected in the floorplate and at the horizontal myoseptum. a–l** Whole mount in situ hybridization for *adamts14*. **a–d** At 26 hpf, *adamts14* transcripts are detectable in the floorplate (black arrow), in cells along the ventral aspect of the segment boundaries (white arrow) and in cells at the HM (black arrowhead). **d** Zoom-in of the squared region in **c**. **e–h** At 36 hpf, *adamts14* expression is visible in the floorplate (black arrow), in cells located at the level of the axial blood vessels (white arrow) and in cells at the HM (black arrowhead). **h** Zoom-in of the indicated region in **g**. **i–l** Expression of *adamts14* is still apparent in the floorplate (black arrow) at 48 hpf as well as in cells at the HM (black arrowhead). **l** Enlargement of the boxed region in **k**. **m**, **n** *adamts14* RNA granules can be detected in cells located in the HM region by RNAscope at 48 hpf. **n** Overlay of the bright field and confocal image for the indicated region in **m**, showing *adamts14* RNA granules in green. **o** Schematic cross view of the trunk illustrating the *adamts14* expression domains (yellow). Scale bars in **a**, **e**, **i**: 100 μm; **b**, **c**, **f**, **g**, **j**, **k**, **m**: 50 μm. DA dorsal aorta, hpf hours post fertilization, HM horizontal myoseptum, M muscle, N notochord, PCV posterior cardinal vein, PL parachordal lymphangioblast.

non-neuronal *adamts3*-expressing cell population, suggesting at least a partial overlap between *adamts3* and *ccbe1* expression. Although *vegfc* expression, based on ISH, has thus far only been reported in arterial and hypochord cells during secondary sprouting stages[31,32], we checked *vegfc* expression levels in *adamts3*+ cell populations and found that non-neuronal *adamts3*+ cells also showed an 3.6-fold *vegfc* mRNA enrichment. These unexpected results suggest that several lymphangiogenic factors, including Vegfc itself, are co-expressed by non-neuronal *adamts3*-expressing cells that are localized close to the migration route of venous and lymphatic secondary sprouts.

Given the importance of any additional expression domains of *vegfc* and *ccbe1* for understanding the temporal-spatial control of localized Vegfc processing, we generated a series of transgenic reporter lines, that would possibly enable us to address co-expression of *adamts3* with other important lymphangiogenic players at single cell resolution. For *vegfc*, expression in the hypochord, the DA and intersegmental arteries has been reported previously[31,32], all expression domains that are relatively distant to the destination of migrating lymphatic precursors. We generated *Tg(vegfc:Gal4FF)* and *Tg(vegfc:mCitrine)* BAC reporter lines, confirming *vegfc* expression in both hypochord and arterial cells. In addition, we found weak expression in individual spinal cord neurons, but also observed a strong signal in the HM region (Fig. 7c). Careful examination revealed the presence of individual mesenchymal cells lining the HM, in intimate contact to PL cells at 48 hpf (Fig. 7c–f). To verify this additional *vegfc* expression domain, we performed ISH using milder proteinase K treatment

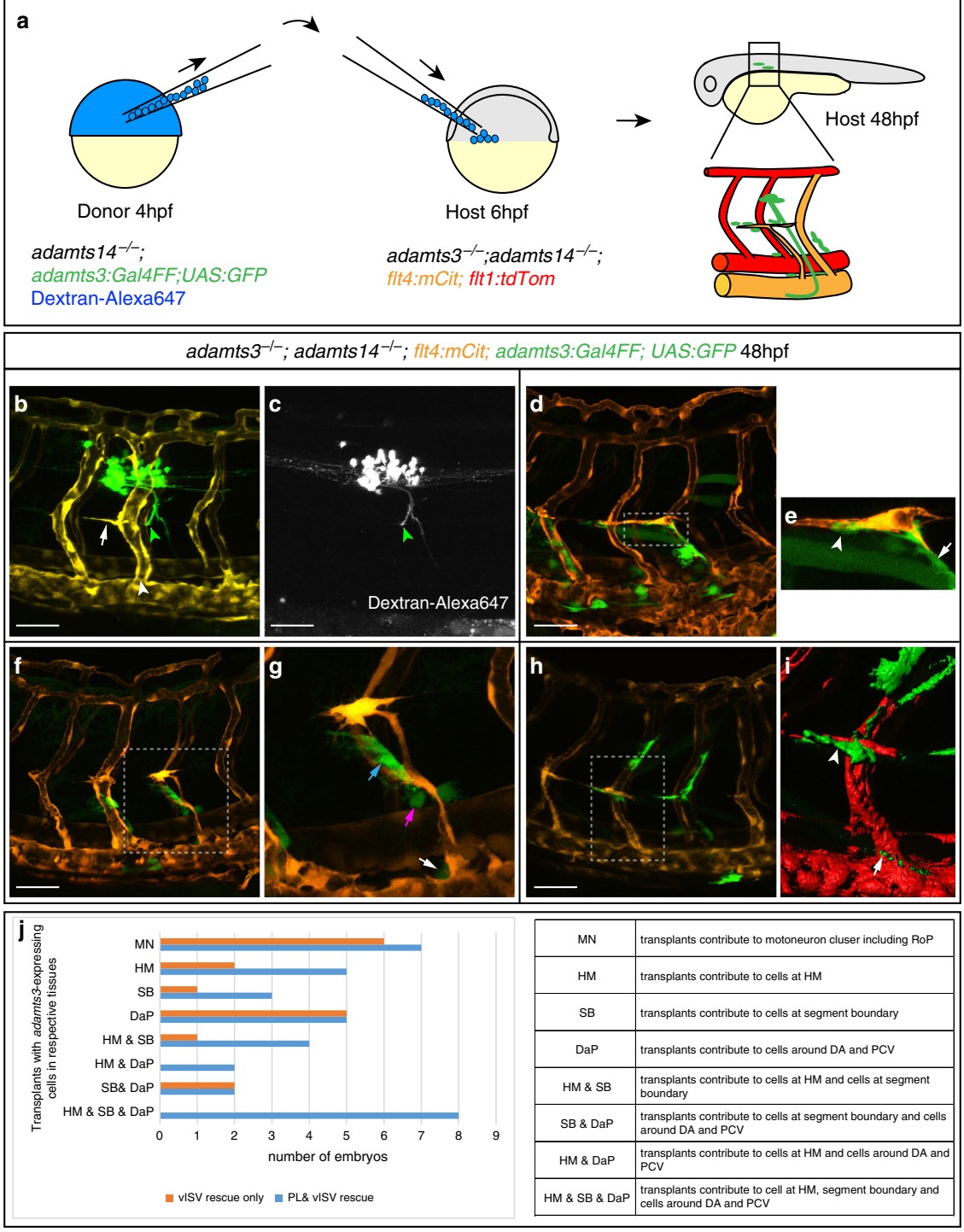

compared with our standard protocol and uncovered a signal at the HM, confirming the expression of *vegfc* in this region of the embryo (Supplementary Fig. 4). We furthermore generated a *Tg (ccbe1:mCit)* BAC reporter and, in addition to the prominent expression in the pronephric duct and the epiphysis (Fig. 7g), we also observed expression in cells at the segment boundaries and the HM (Fig. 7h, i), consistent with previous ISH data[6]. Again, in segments where PL cells were present, they were always in direct contact with *ccbe1*+ cells at the HM (Fig. 7i, j).

We employed these reporter tools to investigate whether *adamts3*, *vegfc*, and *ccbe1* are co-expressed within the mesenchymal

cells at the HM. Substantiating the RNAseq results, we found some *vegfc*-expressing cells at the HM to also express *adamts3* (Fig. 7k, l). Similarly, we observed co-expression of *adamts3* and *ccbe1* transgenes at the midline (Fig. 7m, n) and consequently also overlapping expression of *vegfc* and *ccbe1* (Fig. 7o, p). Compared with UAS:GFP, the UAS:RFP reporter line would mark the respective cells at the HM only in a mosaic fashion for all employed Gal4FF reporter lines, likely due to partial silencing of the UAS elements[33,34]. Hence it remains open whether the overlap between the mesenchymal cells expressing *adamts3*, *ccbe1*, or *vegfc* would be partial or possibly complete.

**Fig. 5 *adamts3* expression via motoneurons or mesenchymal cells can rescue venous sprouting and PL formation in *adamts3; adamts14* double mutants. a** Diagram depicting the heterochronic transplantation approach: Dextran-Alexa647 is injected into one-cell stage *adamts14⁻/⁻; adamts3:Gal4FF; UAS:GFP* embryos (donor embryos). Cells are transferred from donor embryos to *adamts3; adamts14* double mutants at 6 hpf (host embryo) and the effects are assessed at 48 hpf. **b, c** In cases where transplanted cells contributed to *adamts3*-expressing motoneuron clusters (green arrowhead), venous ISVs (white arrowhead), and PL cells (white arrow) formed in double mutants. **c** Transplanted cells labeled by Dextran-Alexa647. **d** When transplants gave rise to *adamts3+* cells at three locations, the HM, the ventral segment boundaries, and close to the DA and PCV, a local rescue of the formation of PL cells and intersegmental veins in *adamts3; adamts14* double mutant embryos was visible. **e** Zoom-in of the boxed region in (**d**) with *adamts3*-expressing cells at the segment boundary highlighted by an arrow and transplanted cells at the HM marked by an arrowhead. **f, g** Transplantation of cells contributing to *adamts3*-expressing cells at the ventral segment boundaries (blue arrow) and to cells located close to the DA (magenta arrow) and the PCV (white arrow) also resulted in a locally restricted rescue of PL and vISV formation. **g** Zoom-in of the indicated region in **f. h, i** Cases, in which the transplants gave rise to *adamts3+* cells at the HM (arrowhead in **i**) and the region around the PCV (arrow in **i**), but not to the cells along the ventral segment boundaries showed a rescue of PL cells and vISVs. **j** Overview about the results obtained from 23 rounds of cell transplantations. Each round contained 144 recipient embryos obtained from an *adamts3+/⁻; adamts14⁻/⁻* incross of which a quarter was expected to be double mutant (in total 828). Embryos showing a rescue of PL or vISV development were selected for imaging and genotyping. All embryos shown in (**j**) are *adamts3; adamts14* double mutants. Scale bars: 50 μm. DA dorsal aorta, hpf hours post fertilization, PCV posterior cardinal vein, PL parachordal lymphangioblast, vISV venous intersegmental vessel.

**The *adamts3+*, *ccbe1+*, and *vegfc+* HM cells co-express *pdgfra*.** The identification of mesenchymal cells expressing all known factors involved in the critical N-terminal processing of Vegfc prompted us to characterize the nature of these cells further. Based on their location and morphology, we excluded them to represent skeletal muscle or neuronal cells, and reasoned that these cells might reflect a fibroblast population. As *pdgfra* is considered a reliable marker for fibroblasts[35], we generated a *Tg(pdgfra:mCit)* transgenic reporter line. At 48 hpf, *pdgfra*-expressing cells were visible throughout the embryo (Fig. 8a, b), particularly populating the HM (Fig. 8c–e) and the segment boundary regions (Fig. 8f). Furthermore, additional *pdgfra+* cells were found juxtaposed to axial and intersegmental blood vessels (Fig. 8f, g). At the HM, *pdgfra*-expressing cells were positioned close to PLs, comparable with the situation for *adamts3*, *vegfc* and *ccbe1*-expressing cells (Fig. 8c–e). Indeed, all *adamts3*-positive cells in the region were also positive for the *pdgfra* reporter, indicating that a subpopulation of *pdgfra*-positive cells provides Adamts3 at the HM (Fig. 8h–m). Analogous co-expression in mesenchymal cells at the HM was found for *vegfc* and *pdgfra* reporters (Fig. 8n–s) and *ccbe1* and *pdgfra* transgenes(Fig. 8t–y). We noticed that signal intensities for all analyzed transgenes varied between individual cells at the HM, suggesting the actual expression levels of the respective genes to be heterogeneous among the mesenchymal cell population. Taken together, these results suggest that the *adamts3*, *ccbe1*, and *vegfc*-expressing mesenchymal cells at the HM reflect a subpopulation of *pdgfra*-positive fibroblasts.

Given the lack of an *adamts14* reporter line, we turned to RNAscope[26], a technique that allows the simultaneous visualization of RNA granules for up to three different genes. We confirmed the expression of *adamts14*, *pdgfra*, *ccbe1*, and *vegfc* at the HM (Supplementary Figs. 5–7). However, dense packing and irregular shapes of cells in this part of the embryo hampered a reliable assessment of whether individual cytoplasmic RNA granules resided within the same or in neighboring cells. We therefore restricted co-expression analysis to individual confocal sections focusing on RNA granules that overlapped with individual DAPI signals and that therefore should reside within the same nucleus and hence the same cell. Following this strategy, we confirmed co-expression of *vegfc* and *pdgfra*, of *ccbe1* and *pdgfra* as well as co-expression of *vegfc* and *ccbe1* (Supplementary Figs. 5 and 6a). We also found cells that co-expressed *adamts14* with *vegfc*, *ccbe1*, or *pdgfra* mRNA (Supplementary Figs. 6b and 7), suggesting that *adamts14* is transcribed in at least part of the *pdgfra*-positive HM fibroblasts. Due to the aforementioned technical difficulties, however, we were unable to quantitatively assess the extent of co-expression between the individual Vegfc-processing components using this technique.

**Mesenchymal cells at the HM represent a fibroblast subtype.** To characterize the *pdgfra+* mesenchymal cell population further and to firmly establish their identity, we carried out single-cell RNA sequencing (scRNAseq). While *adamts3* and *ccbe1* are not only expressed in HM cells but also in cells at the segment boundaries and around the DA and PCV, we took advantage of the rather confined and strong expression of the *vegfc:Gal4FF; UAS:GFP* transgene in those cells at the HM, along with arterial ECs and hypochord cells at 48 hpf (Fig. 7c–e). We furthermore took into consideration that the expression of *adamts3*, *adamts14*, *ccbe1*, and *vegfc* might occur in a salt-and-pepper-like pattern, resulting in some cells expressing one particular combination of those genes while others might express a different combination. Hence, we used the *pdgfra* transgene to additionally sort for a broader fibroblast population. Employing either *vegfc:Gal4FF; UAS:GFP;kdrl:mCherry* double transgenic embryos or embryos harboring the *pdgfra:mCit* reporter, we established a partial dissociation protocol in which the trunks were only superficially dissociated. We thus enriched for peripheral cells (such as cells positioned at the HM) while reducing the respective representation of more medial cells (e.g., DA or hypochord cells) in the cell suspension (Fig. 9a) and FAC-sorted in total three different cell populations that were subsequently subjected to single cell sequencing: (1) *vegfc:Gal4FF;UAS:GFP +*; *kdrl:mCherry-* cells, (2) *kdrl:mCherry+*; *vegfc:Gal4FF;UAS:GFP-* cells, and (3) *pdgfra: mCit+* cells (Supplementary Fig. 3b, c).

A total of 1109 single cell transcriptomes from all three cell populations was subsequently divided into 24 Pagoda2 clusters (Fig. 9b, Supplementary Fig. 8a, b). Guided by canonical cell-class marker expressions, we identified an endothelial cell cluster (cluster 20) that consisted exclusively of *kdrl:mCherry+* cells and that expressed markers such as *pecam1*, *dll4*, *cdh5*, *fli1a*, and *fli1b* (Supplementary Figs. 8a–c and 10). As expected from the general expression pattern of the transgenic lines we employed, we furthermore identified clusters representing immune cells in cluster 5 (*spi1b+*, *mpeg1.1+*) and cluster 18 (*mpx+*, *lyz+*), neuronal cells in cluster 7 (*gfap+*, *npas1+*) and cluster 15 (*elav3+*, *elav4+*), hypochord cells in cluster 23 (*col8a1a+*, *col2a1a+*, *fxyd1+*, *foxa1+*, *matn3b+*) and epithelial cells in cluster 16 and 17 (*cdh1+*, *cldnb+*, *cdh17+*, *epcam+*), likely representing pronephric duct cells (Supplementary Figs. 8b, c and 10).

Significantly, we found that 75% of the isolated cells grouped into clusters that can be categorized as a general fibroblast population, based on expression of classical fibroblast markers such as *pdgfra*, *col1a1a*, *lum*, and *dcn* (Fig. 9c–f, Supplementary Fig. 8b, c). Amongst those fibroblast clusters we noticed strong enrichment for *vegfc* and *ccbe1* transcripts within cluster 2, and to

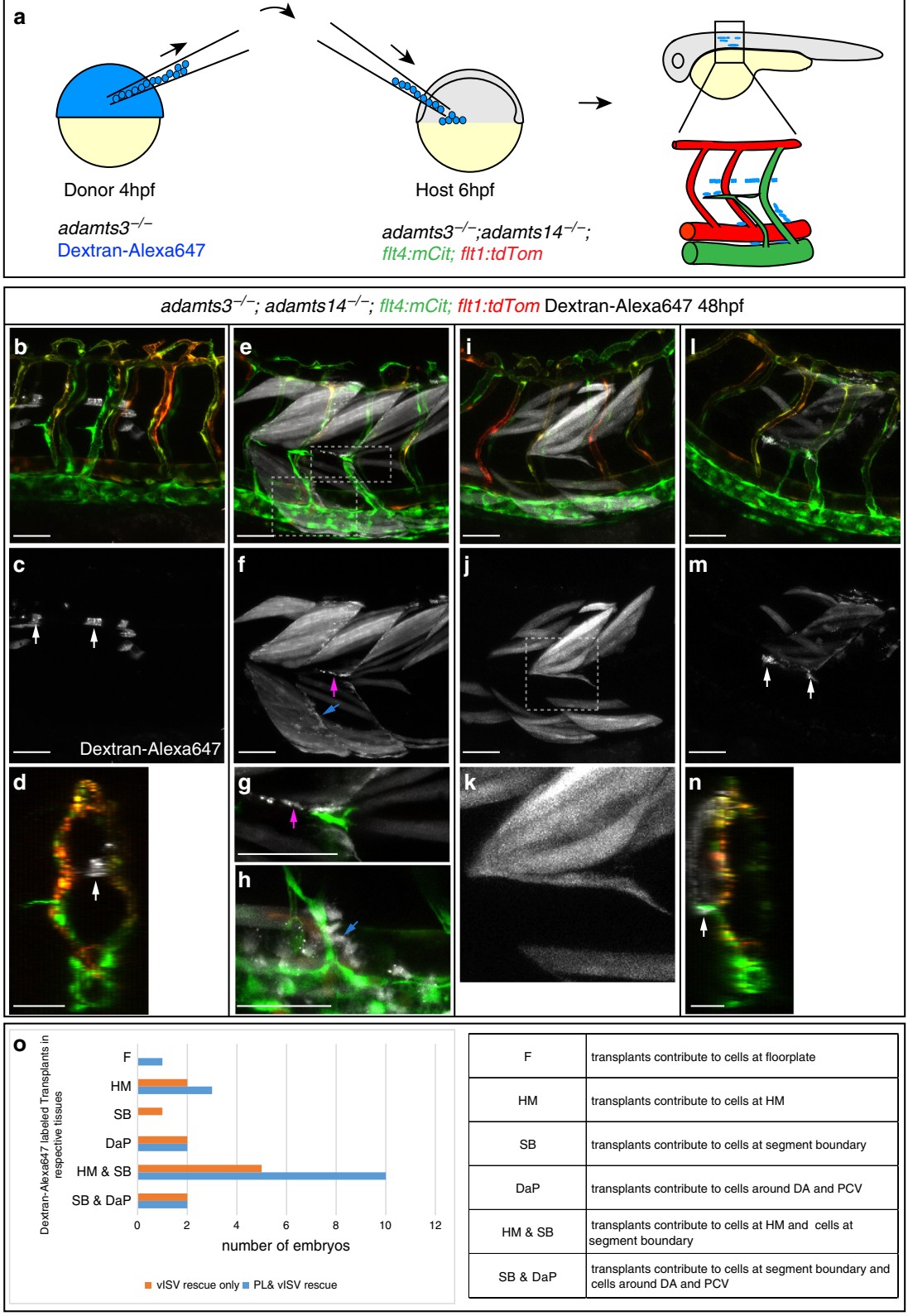

some extent in clusters 1 and 4 (Fig. 9g, h). Given high expression levels of genes related to cell proliferation (*mki67*, *cdk1*, *top2a*, *ccna2*), we concluded that cluster 4 contains proliferating cells (Supplementary Fig. 8b, c). Cluster 1 on the other hand seemed to be related to cluster 2 and might represent a precursor state of the

more differentiated fibroblast population in cluster 2, as suggested by velocity cell-differentiation prediction analysis (Supplementary Fig. 9).

In order to narrow down which fibroblast cluster represents the mesenchymal cell population expressing lymphangiogenic genes

**Fig. 6 *adamts14*-expressing floorplate and mesenchymal cells can rescue venous sprouting and PL formation in *adamts3; adamts14* double mutants.**
**a** Scheme outlining the transplantation assay: embryos homozygous mutant for *adamts3* were injected with Dextran-Alexa647 to mark the donor cells. At 4 hpf, cells from the donor were transplanted into *adamts3; adamts14* double mutant host embryos (6 hpf). Host embryos were subsequently analyzed for rescue of venous sprouting and the location of the transplanted material at 48 hpf. **b**–**d** Transplantation of cells contributing to the floorplate (arrow) resulted in the formation of intersegmental veins and venous sprouts reaching the HM region. **c** Dextran-Alexa647 labeled transplanted tissue. **d** Virtual cross section showing the transplanted floorplate cells (white) in the rescued area. **e**–**h** In cases where the transplanted cells contributed to muscle but also to individual cells at the HM (magenta arrow in **f**, **g**), at segment boundaries (blue arrow in **f**) and also to cells located directly adjacent to the DA and PCV (blue arrow in **h**), PL formation and vISV formation was locally restored in *adamts3; adamts14* double mutants. **f** Dextran-Alexa647 channel highlighting the transplanted tissue. **g**, **h** Partial z-projections of the boxed regions in (**e**) indicating transplanted cells in white. **i**–**k** Transplants giving only rise to muscle tissue did not rescue the venous sprouting defects in double mutants. **j** Dextran-Alexa647 labeled transplanted tissue. **k** Zoom-in of the boxed HM region in **j**. **l**–**n** Cell transplantations giving rise to tissues including cells located at the HM (arrow in **m**) resulted in a rescue of PL and vISV formation in this area at 48 hpf. **n** Virtual cross section indicating the position of the transplanted cell at the HM (arrow). **o** Graph summarizing the results from 13 rounds of cell transplantations. Each round contained 144 recipient embryos derived from an *adamts3*[+/−]; *adamts14*[−/−] incross. From the expected 468 double mutant recipients, embryos showing PL or vISV rescue were selected for imaging and genotyping. All embryos shown in (**o**) are *adamts3; adamts14* double mutants. Scale bars: 50 μm. DA dorsal aorta, HM horizontal myoseptum, hpf hours post fertilization, PCV posterior cardinal vein, PL parachordal lymphangioblast, vISV venous intersegmental vessel.

at the HM, we checked which transcripts would be specifically highly expressed within cluster 2 when compared (a) with all other clusters, (b) to all fibroblast clusters, or (c) to the closely related progenitor cluster 1 (Supplementary Tables 1–3). Amongst the most differentially expressed genes we noticed *engrailed 1a* and *engrailed 1b* whose expression seems largely confined to cluster 2 and to a lesser extent to some cells within clusters 1 and 4 (Fig. 9i, j). Comparison between the *vegfc*, *ccbe1* and *en1a*, *en1b* t-SNE plots demonstrated a high degree of overlapping expression (Fig. 9g–j). We therefore performed antibody staining at 48 hpf to check whether Engrailed expression marks the cells of interest at the HM. Engrailed has been used as a marker for muscle pioneer cells, which can be distinguished from 13 hpf onwards and are located at the midline[36]. These 2–6 cells per somite represent a form of slow muscle fibers and are the first muscle cells that differentiate[36,37]. We found that the Engrailed antibody, which cross-reacts with zebrafish *en1a*, *en1b*, *en2a*, and *en2b*[38], showed restricted staining of nuclei in the HM region at 48 hpf. Importantly, all *vegfc:Gal4FF;UAS:GFP* positive cells at the HM were also co-stained by Engrailed, but Engrailed also highlighted other nuclei in the region (Supplementary Fig. 11a–c), indicating that not only fibroblast populations express Engrailed proteins at this stage of development in the midline area. Antibody staining on *pdgfra:mCit* (Supplementary Fig. 11d–f) or *adamts3:Gal4FF;UAS:GFP* embryos (Supplementary Fig. 11g–i) revealed that *pdgfra*+ or *adamts3*+ cells at the HM, but not in other embryonic regions, were co-stained by Engrailed, demonstrating that the overlapping expression of *en1a/en1b* with *vegfc* or *pdgfra* can be used as a specific landmark for the fibroblast subpopulation at the HM.

After verifying that cluster 2 constitutes the fibroblast subpopulation in question, we also checked expression of other genes involved in lymphatic development in cells within this cluster. The number of *adamts3*+ cells within cluster 2 was much lower as compared with *vegfc* and *ccbe1*, indicating that the protease would be secreted in a salt-and-pepper-like fashion by only a fraction of HM fibroblasts at this stage (Fig. 9k). In addition, several *adamts3*-expressing cells were found within the neuronal clusters and within other fibroblast clusters, likely reflecting the expression domains along segment boundaries and around axial vessels. Finally, FAC-sorted cell populations contained only very few cells with active *adamts14* transcription, that were mostly found in cluster 2 and the related cluster 1 (Fig. 9l), suggesting the fraction of *pdgfra* + fibroblasts secreting Adamts14 at the HM to be relatively small.

Two other important genes regulating the migration of lymphatic precursor cells are *cxcl12a* and *svep1*, both of which

have been reported to be expressed by unknown cells located at the HM[39–41]. In our scRNAseq data, *cxcl12a*-expressing cells showed strong enrichment in cluster 2, very similar to *vegfc* and *ccbe1* (Fig. 9m). The *svep1* gene has been reported to be expressed at regions with venous and LEC migration activity, including the HM. In line with the notion that only relatively few cells express *svep1* at the HM[39], we found that only a fraction of the fibroblast cells within cluster 2 actively express *svep1* at 48 hpf (Fig. 9n). These results demonstrate that also additional regulators of lymphatic development are provided by fibroblast subpopulations located at the HM, emphasizing their importance as local signaling hubs within the embryonic trunk.

Our co-expression analysis using transgenic embryos (Fig. 7) demonstrated that the expression of *adamts3*, *ccbe1*, and *vegfc* at least partially overlaps at the HM, and that all cells expressing one of these genes in this region would always be *pdgfra*-positive. In order to assess whether a single *pdgfra*+ fibroblast at the HM would likely express more than one of the genes involved in Vegfc processing, we analyzed the co-expression of the respective transcripts in cluster 2 cells in more detail (Fig. 9o–s, Supplementary Table 4). 80% (47/59) of the actively *vegfc*-transcribing cells also contained *ccbe1* transcripts, while 47% (47/99) of *ccbe1*-expressing cells also transcribed *vegfc*, indicating that both genes have a high tendency to be co-expressed (although more fibroblasts express *ccbe1* (72%) compared with *vegfc* (43%) at 48 hpf). The proportion of cells within cluster 2 actively expressing *adamts3* was 9% (13/137) and for *adamts14* 3% (4/137). While we did not observe co-expression of *adamts3* and a*damts14*, we noticed that 85% (11/13) of *adamts3*-expressing cells co-expressed *ccbe1* and that the overlap between *adamts3* and *vegfc* was 38% (5/13). Interestingly, we also identified fibroblast cells in cluster 2 expressing one of the two protease genes together with *vegfc* and *ccbe1* simultaneously (4/13 of *adamts3*+ or 1/4 of *adamts14*+ cells). Taken together, these results indicate that although HM fibroblasts have a high tendency to co-express several of the genes in question, the proportion of cells expressing a particular gene at the HM at 48 hpf differs significantly, suggesting a certain heterogeneity amongst this fibroblast population.

**Guidance of PL cells requires Vegfc processing at the HM.**
Since it has never been functionally addressed whether Vegfc processing at the HM is required to recruit PLs to this region, we assessed at which of the *vegfc* expression domains processing needs to occur in order to allow for the formation of vISVs or PLs, respectively. To this end, we locally expressed fully processed, mature human VEGFC in mutants with disabled Vegfc

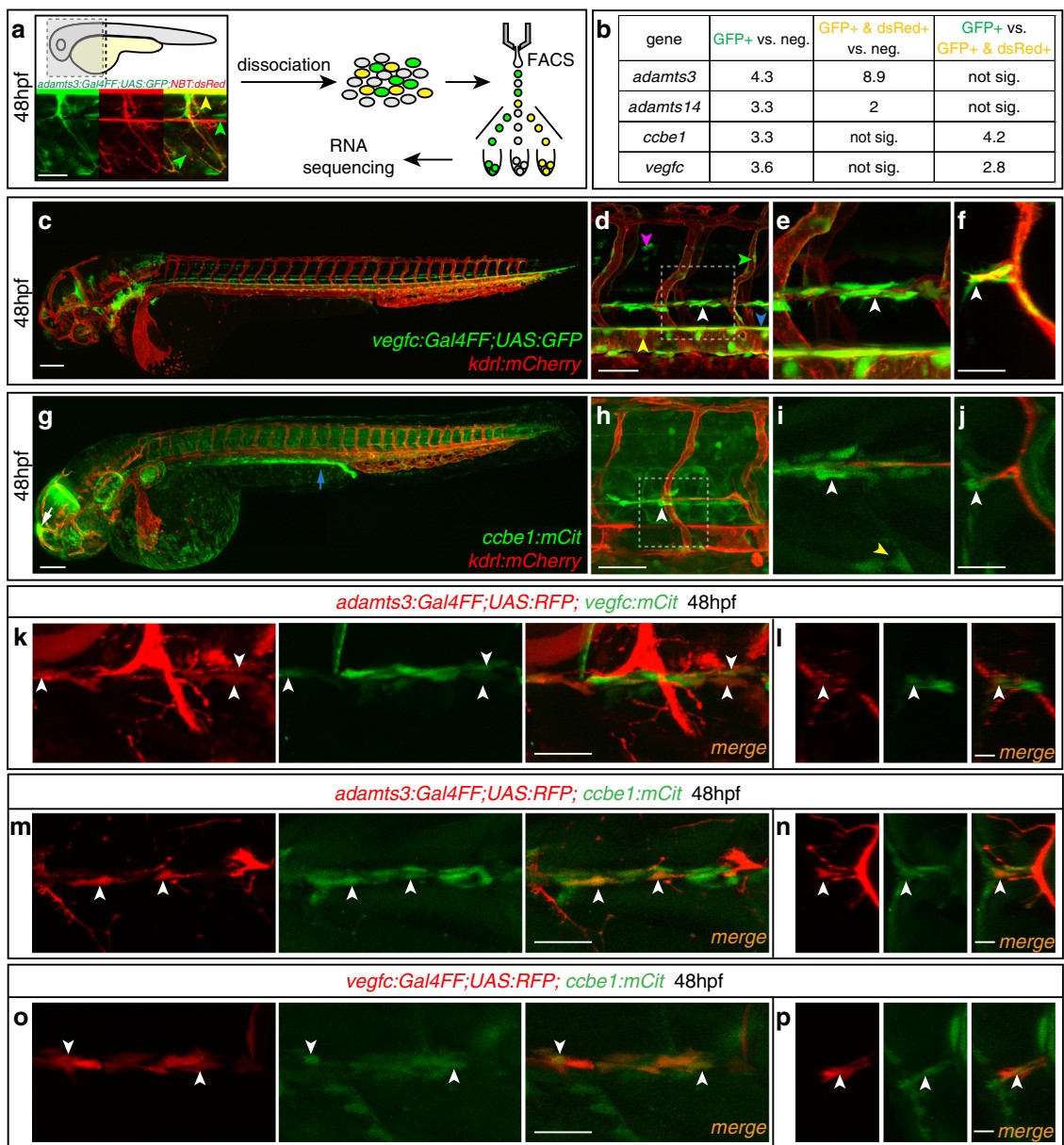

**Fig. 7 Non-neuronal *adamts3*-expressing cells co-express *vegfc* and *ccbe1*. a** Strategy for transcript analysis of *adamts3*-expressing cell populations by RNA sequencing. *adamts3:Gal4FF;UAS:GFP*; *NBT:dsRed*-positive trunks were dissociated at 48 hpf and cells were FAC-sorted into (1) *adamts3*-expressing neurons (GFP +/dsRed+, yellow arrowhead), (2) *adamts3* + non-neuronal cells (GFP+, green arrowheads) and (3) a negative control. Pooled cell populations were analyzed by RNA sequencing. **b** Summary of RNA sequencing analysis. Values reflect fold-changes in transcript abundance for indicated genes upon comparison of cell populations stated in the column headings. To identify significant gene expression differences, *p* values were determined by Wald test and an FDR < 0.05 and FDR-adjusted *p* value < 0.1 cutoff was used (source data are provided as a Source Data file). **c** Lateral view of a *vegfc: Gal4FF; UAS:GFP; kdrl:mCherry* transgenic embryo at 48 hpf. **d** Higher magnification showing *vegfc* expression in HM cells (white arrowhead), in the hypochord (blue arrowhead), the DA (yellow arrowhead), in arteries (green arrowhead) and in neurons (magenta arrowhead). **e** Zoom-in of the boxed region in (**d**) highlighting *vegfc*-expressing cells at the HM (arrowhead). **f** Virtual cross section focusing on the HM, showing juxtaposition between a PL and *vegfc*-expressing cells (arrowhead). **g** Lateral view of a *ccbe1:mCitrine; kdrl:mCherry* transgenic embryo at 48 hpf showing *ccbe1* expression in the pronephros (blue arrow) and epiphysis (white arrow). **h** Higher magnification demonstrating *ccbe1* expression at the HM (arrowhead). **i** Partial projection of the marked region in (**h**) showing *ccbe1*-expressing cells at the HM (white arrowhead) and the segment boundary (yellow arrowhead). **j** Virtual cross section indicating *ccbe1*+ cells at the HM (arrowhead). **k**, **m**, **o** Lateral views (partial z-projections) and **l**, **n**, **p** virtual cross sections of the HM region at 48 hpf. Arrowheads indicate cells with co-expression of the indicated transgenes. **k**, **l** *adamts3* and *vegfc* are co-expressed by individual cells at the HM. **m**, **n** Mesenchymal cells at the HM co-express *adamts3* and *ccbe1*. **o**, **p** Overlapping expression of *vegfc* and *ccbe1* in HM cells. Scale bars in **a**, **d**, **h**: 50 μm; **c**, **g**: 100 μm; **f**, **j**, **k**, **m**, **o** 25 μm; **l**, **n**, **p**: 10 μm. DA dorsal aorta, HM horizontal myoseptum, hpf hours post fertilization, neg. negative cell population, sig. significant.

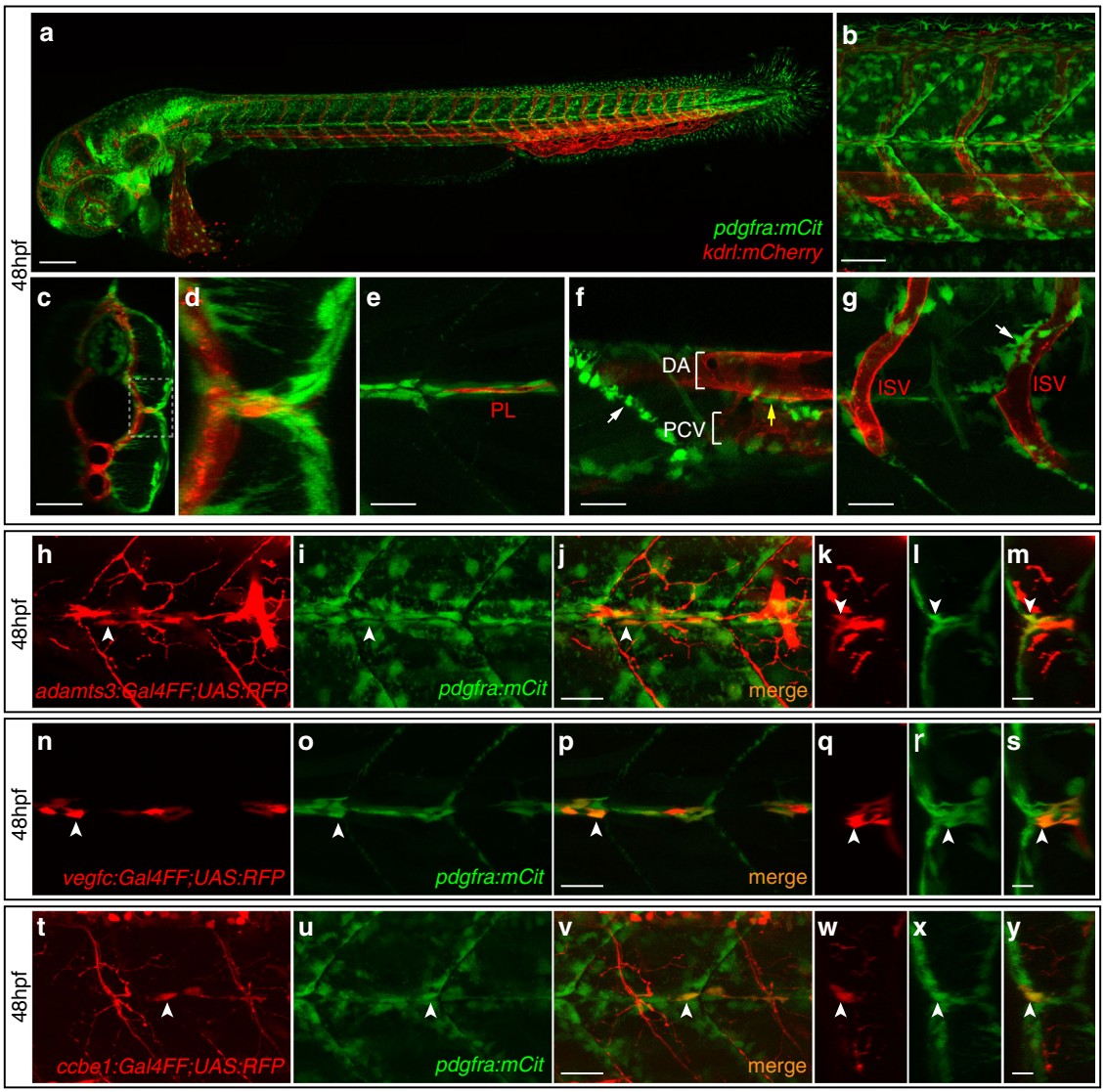

**Fig. 8 Co-expression of *adamts3*, *ccbe1*, and *vegfc* with *pdgfra* at the horizontal myoseptum. a–g** Confocal images of *pdgfra:mCitrine; kdrl:mCherry* double transgenic embryos at 48 hpf. **a** Lateral view giving an overview of the *pdgfra*-expression domains. **b** Higher magnification of the trunk region above the yolk extension. **c**, **d** Virtual cross section of the trunk shown in **b**. **d** Zoom-in of the boxed area in (**c**) showing the HM region in detail. **e–g** Partial projections of (**b**). **e** At the HM, *pdgfra*-expressing cells (in green) are surrounding the PL cells (in red). **f** *pdgfra* is expressed by cells located along the segment boundaries (white arrow) as well as by individual cells located around the main axial blood vessels (yellow arrow). **g** At the level of the ISVs, individual *pdgfra+* cells locate in close proximity to the blood vessels. **h–j**, **n–p**, and **t–v** Lateral views and **k–m**, **q–s**, and **w–y** virtual cross sections of the HM region in double transgenic embryos at 48 hpf. White arrows highlight cells with co-expression. **h–m** A subpopulation of *pdgfra*-expressing cells (green) is co-labeled by the *adamts3* reporter transgene (red) at the HM. **n–s** Cells at the HM co-express *vegfc* (red) and *pdgfra* (green). **t–y** All *ccbe1*-expressing cells at the HM (red) show co-expression of *pdgfra* (green). Scale bar in **a**: 100 μm; **b**, **c**: 50 μm; **e–j**, **n–p**, **t–v**: 25 μm; **k–m**, **q–s**, **w–y**: 10 μm. DA dorsal aorta, HM horizontal myoseptum, hpf hours post fertilization, ISV intersegmental vessel, PCV posterior cardinal vein, PL parachordal lymphangioblast.

processing: injection of a *UAS:ΔNΔC-VEGFC-P2A-RFP* construct into embryos double mutant for *adamts3* and *adamts14* and harboring the *vegfc:Gal4FF* transgene (Fig. 10a) resulted in mutant embryos that expressed fully mature VEGFC in a mosaic fashion within the normal *vegfc* expression domains. Injected embryos were subsequently analyzed for a rescue in vein and PL formation, taking into account which cells provided processed VEGFC (labeled by RFP expression).

Thus far, it had been assumed that *vegfc* is expressed only by hypochord and DA cells at the time of secondary sprouting, posing the question how Vegfc could instruct PLs to pass this area and progress to the HM. When providing mature VEGFC at the level of the DA by expression in either hypochord or DA cells, we observed sprouting of venous endothelial cells in the immediate

proximity, resulting in the formation of a vISV close by (Fig. 10b–e). In other cases, DA and especially hypochord expression of ΔNΔC-VEGFC not only resulted in the formation of a vein but also induced a local hyper-sprouting phenotype of venous ECs, possibly due to higher expression levels of the construct in these embryos. As a result, excessive venous structures extended around the source of mature VEGFC protein (Fig. 10f–i), indicating that the local levels of mature Vegfc need to be tightly controlled to allow for normal vascular development. A rescue of PL cells, in contrast, was only observed when processed VEGFC was provided by fibroblasts at the HM region, mostly in conjunction with expression within the DA (Fig. 10j–q). These results indicate that controlled processing of endogenous Vegfc at the DA level is important and sufficient to trigger the

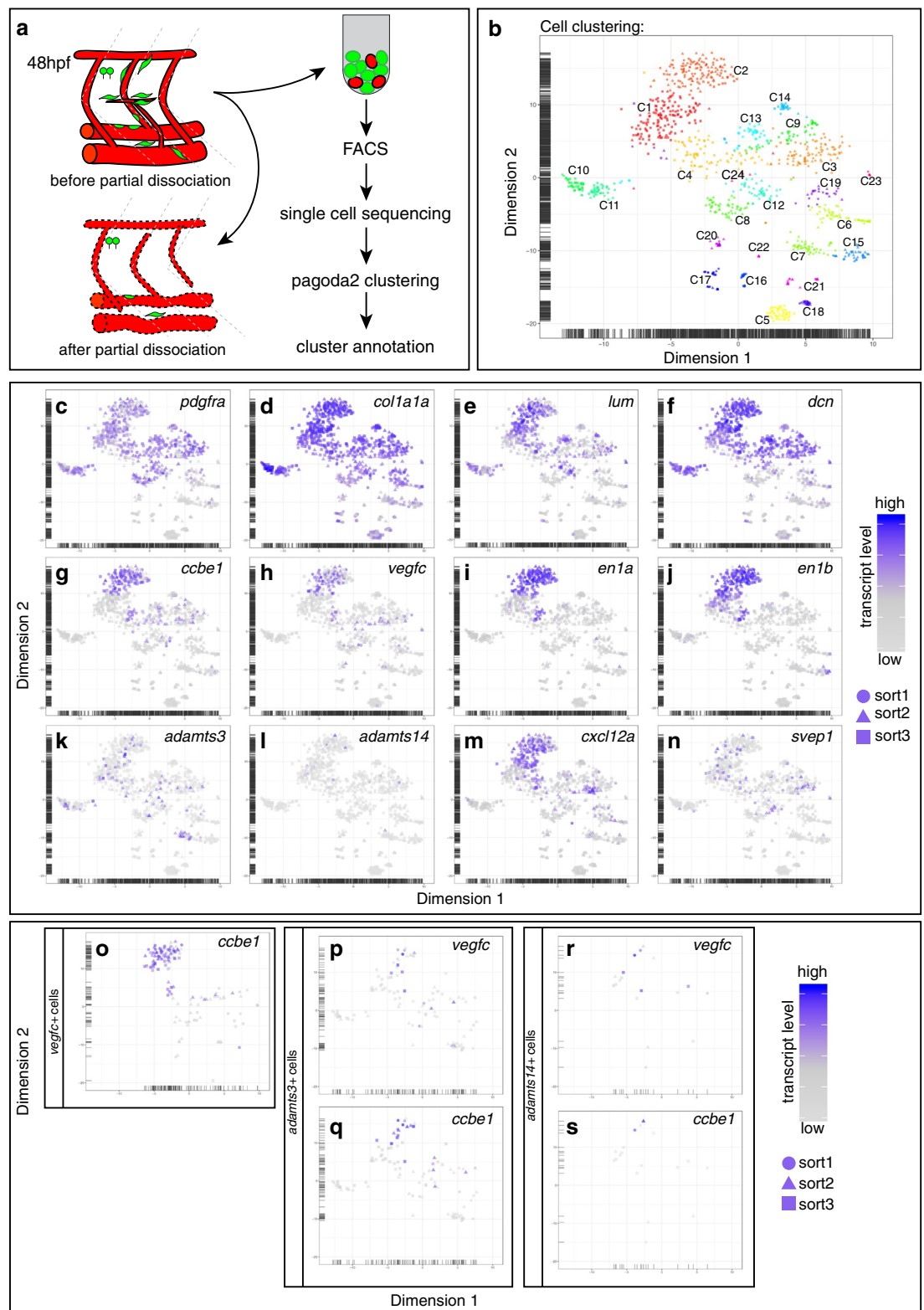

initial sprouting phase from the vein and the formation of vISVs. In order to allow for formation of PLs, however, a source of mature Vegfc at the HM region appears essential. We noticed that in cases of PL rescue, lymphatic precursor cells would always end up in direct proximity to the ΔNΔC-VEGFC-expressing fibro-blasts, indicating that the sprouts indeed migrated toward the source of mature VEGFC. The effects elicited by processed VEGFC appeared locally restricted in all cases, suggesting that the

mature protein does not diffuse very far into neighboring segments of the embryo.

## Discussion

Here we assessed the expression and interplay of Adamts3 and Adamts14 with Ccbe1 in N-terminal processing of secreted Pro-Vegfc and hence the activation of the Vegfr3/Flt4 signaling pathway in the spatio-temporal regulation of lymphangiogenesis.

**Fig. 9 Single-cell RNA sequencing identifies fibroblast subpopulations expressing *adamst3, adamts14, ccbe1,* and *vegfc*. a** Scheme of the partial dissociation approach and the work flow. Note that embryonic trunks were not completely dissociated in order to obtain a higher proportion of cells from more superficial tissues while reducing the amount of cells in the suspension originating from deeper tissues. **b** Pagoda2 clustering of the individual cell transcriptomes gives rise to 24 clusters (labeled as C1-C24). **c**–**n** t-SNE plots indicating the transcript levels of the respective gene in each sequenced cell. Expression of the general fibroblast markers *pdgfra* (**c**), *col1a1a* (**d**), *lum* (**e**), and *dcn* (**f**). Expression of *ccbe1* (**g**) and *vegfc* (**h**) is largely restricted to cells within cluster 2 as is the case for the expression of the genes *en1a* (**i**) and *en1b* (**j**). *adamts3-* (**k**) and *adamts14*-expressing cells (**l**) are distributed over several clusters. **m**, **n** While *cxcl12a*-expressing cells are enriched in cluster 2, *svep1*-expressing cells can be found within several different clusters. t-SNE plots highlighting cells co-expressing *vegfc* and *ccbe1* (**o**), *adamts3* and *vegfc* (**p**), or *adamts3* and *ccbe1* (**q**) as well as cells co-expressing *adamts14* and *vegfc* (**r**) or *adamts14* and *ccbe1* (**s**). Symbol shapes in the t-SNE plots indicate from which FACS run the cells had been obtained (circles and triangles originate from sorts using *vegfc:Gal4FF; UAS:GFP; kdrl:mCherry* embryos, cells indicated by squares stem from the FAC-sorting of *pdgfra:mCitrine* embryos), color scales reflect the relative transcript levels. hpf hours post fertilization.

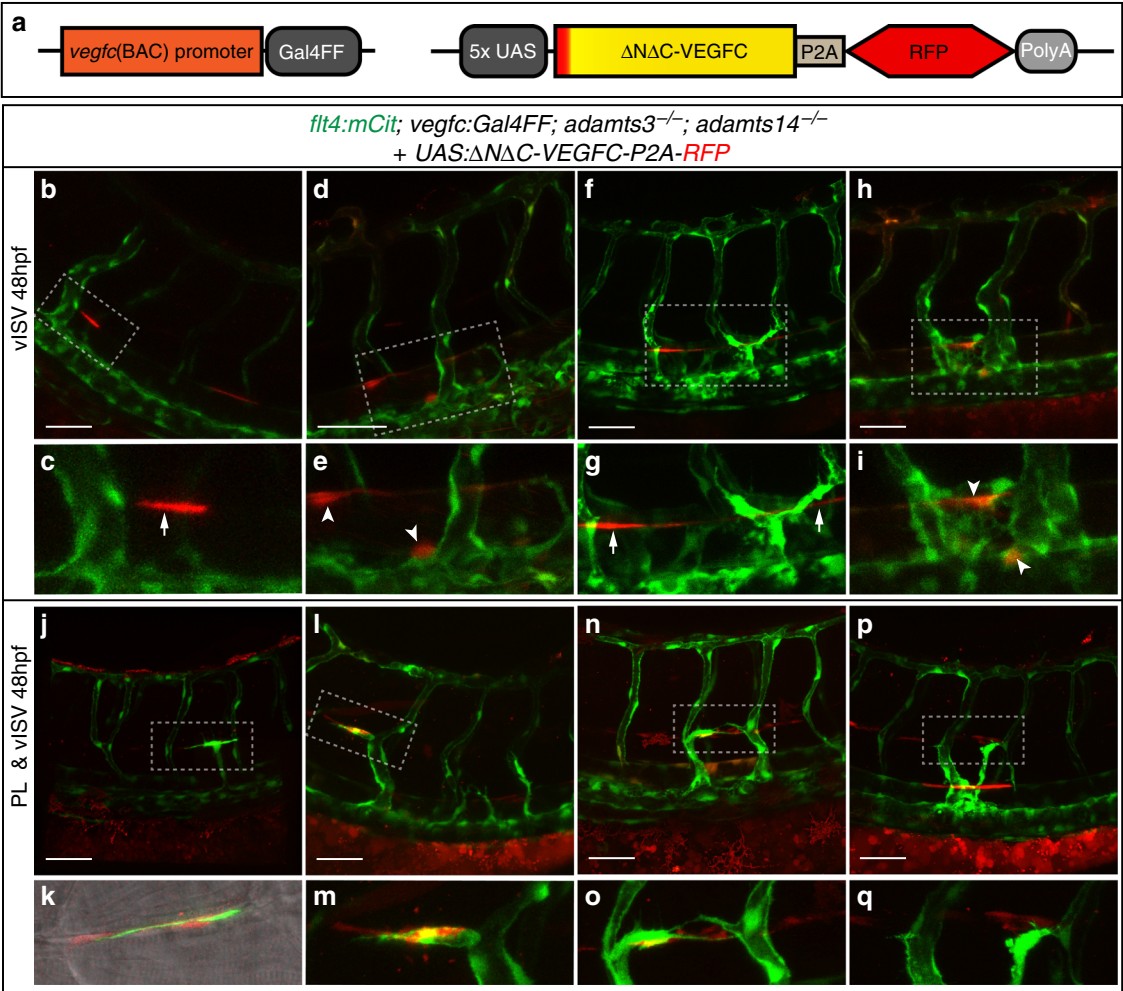

**Fig. 10 Local Vegfc processing at the HM is required to guide endothelial cells to this region. a** Schematic depiction of the mature human VEGFC overexpression construct and the *vegfc:Gal4FF* transgenic line, which was used to drive mosaic expression of the construct upon transient injection. **b**–**q** *vegfc: Gal4FF* positive embryos were injected with the overexpression construct at the one-cell stage and imaged at 48 hpf. All embryos shown are *adamts3; adamts14* double mutant, transgenic for *flt4:mCitrine* (green) and express *UAS:ΔNΔC-VEGFC-P2A-RFP* (red). **b, c** Expression of *UAS:ΔNΔC-VEGFC-P2A-RFP* in hypochord cells (white arrow) rescues vein formation. **d, e** Local vISV rescue upon expression of mature VEGFC in DA cells (white arrowhead). **f**–**i** Venous hyper-sprouting with ectopic venous structures formed in the vicinity of hypochord cells (**f, g**, arrows) or DA cells (**h, i**, arrowheads) with expression of the ΔNΔC-VEGFC construct. **j, k** *UAS:ΔNΔC-VEGFC-P2A-RFP* positive fibroblasts recruit PL cells to the HM region. **l, m** Expression of mature VEGFC in hypochord cells and in fibroblasts at the HM rescues vISV and PL formation in *adamts3; adamts14* double mutant embryos. Note the close proximity of the PL cell to the ΔNΔC-VEGFC producing fibroblasts. **n, o** Combined expression of mature VEGFC in DA cells and fibroblasts at the HM results in the formation of vISV and PL cells close by. **p, q** Simultaneous expression of ΔNΔC-VEGFC in hypochord cells and in fibroblasts at the HM results in a local rescue of PL and vISV formation but can additionally lead to a hyper-branching of venous structures at the level of the DA. **c, e, g, i, k, m, o, q**: Zoom-in image of the indicated regions in **b, d, f, h, j, l, n, p**. Scale bars: 50 μm. DA dorsal aorta, HM horizontal myoseptum, hpf hours post fertilization, PL parachordal lymphangioblast, vISV venous intersegmental vessel.

In particular, using single cell sequencing and cell transplantations we resolved how the different cellular sources of the individual players and their spatial positions determine the migratory routes taken by the lymphatic precursor cells, which is relevant for the correct patterning of the lymphatic system.

We surprisingly found that—in contrast to mice and men[11–13]—zebrafish *adamts3* knock-outs fail to display lymphatic defects. Only the simultaneous loss of Adamts3 and Adamts14 activity interferes with lymphatic development, resulting in an exceptionally clean phenotype: *adamts3;adamts14* double mutants fully recapitulate the venous sprouting defects seen in *ccbe1* and *vegfc* mutants[6,10,42], suggesting that in fish both Adamts3 and Adamts14 are able to activate Vegfc. One remaining wild-type copy of either gene is sufficient to trigger lymphangiogenesis, demonstrating that both proteases act redundantly. Zebrafish Adamts14 could also activate human VEGFC in an in vivo activity assay (Fig. 2), which prompted us to revisit the VEGFC cleavage capacity of human ADAMTS14 in vitro. We found that human VEGFC is not only proteolytically processed upon ADAMTS3 addition as described previously[8] but also in case recombinant ADAMTS14 protein is added to the conditioned medium. The processing capacity of both ADAMTS proteases is thus an evolutionary conserved feature. Although ADAMTS3 seems to be the essential VEGFC-processing protease during murine lymphatic sprouting[11,12], future studies will clarify a possible involvement of *Adamts14* during other aspects of lymphatic development.

*adamts3* and *adamts14* exhibit strong expression domains within the spinal cord, but also possess partially overlapping expression domains in fibroblast subpopulations located along the migration route of LECs and the HM. Previous work has indicated a correlation between the presence of specific motoneurons and the migration of lymphatic precursor cells towards the HM, an area to which RoP motoneurons extend axonal projections[30]. These results suggested a model in which motoneurons would be directly involved in guiding migrating PLs, possibly by directed secretion of proteins from their axons, but the exact nature of such a connection remained unclear. Here we report that Adamts3 secreted by motoneurons directly impacts the sprouting and migration of secondary sprouts. Cell transplantations indeed showed that in absence of endogenous Adamts3 or Adamts14, clusters of *adamts3*-expressing motoneurons are sufficient to rescue venous sprouting, the formation of vISVs and even PL migration to the HM (Fig. 5). This establishes a direct connection between embryonic motoneurons and the guidance of lymphatic precursor cells. For *adamts14* we identified persisting expression in the floorplate. In cases where transplanted *adamts14*[+/+] cells contributed to the floorplate in double mutant embryos, a rescue of secondary sprouting could be observed, identifying Adamts14 as a floorplate-derived protein that influences embryonic lymphangiogenesis.

Besides these neuronal expression domains, we identified additional sources for *adamts3* and *adamts14*. A Tg(*adamts3:Gal4FF*) reporter line highlights individual cells located close to the PCV and DA as well as cells aligned along the segment boundaries, reminiscent of a recently described population of sclerotome-derived tendon fibroblasts called tenocytes[43,44]. Furthermore, we found such cells at the HM, where they are positioned in direct contact to the incoming PL cells. Single cell transcriptome analysis identified these cells as a population of *pdgfra*-positive fibroblasts. This fibroblast subpopulation resident at the HM and positive for *en1a/en1b* turned out to express not only *adamts3* and *adamts14* in a salt-and-pepper-like fashion, but also to provide other known key components of the Vegfc-processing machinery such as Ccbe1 and Vegfc itself, which thus far had been believed to be only provided by arterial cells of the DA and ISVs, as well as by hypochord cells that are located

relatively far from the HM region to which lymphatic precursor cells migrate. Employing transgenic reporters we confirm expression domains for *ccbe1* and *vegfc* within the *pdgfra+* fibroblasts at the HM, indicating that this cell population provides all players known to be required for the generation of fully active Vegfc in the extracellular space. Thus, fully processed, mature Vegfc protein is likely produced at the HM, guiding PLs to this region.

In order to firmly establish at which location Vegfc maturation needs to happen to allow for (1) normal sprouting and formation of vISVs versus (2) the migration of lymphatic precursor cells to the HM, we provided fully processed Vegfc in different anatomical compartments in a mosaic fashion. This analysis revealed that processing at the level of the DA/hypochord can trigger sprouting from the PCV and formation of intersegmental veins. Ccbe1 and Adamts3 (and likely also Adamts14) are provided by fibroblasts located close to the axial vessels, suggesting that for this initial phase of migration, these fibroblasts provide the molecular components for Vegfc to mature and to allow formation of vISVs. In an independent step, future PL cells that will go on to populate the HM region seem to depend on a local source of mature Vegfc at the HM, as we only observed normal PL formation in embryos that contained fibroblasts expressing the ΔNΔC-VEGFC construct at the HM. Given the expression of all required factors in *pdgfra+* fibroblasts in this area, the secretion of these factors by the same or neighboring fibroblasts at the HM region would lead to the local production of mature Vegfc, in turn attracting and guiding the lymphatic precursor cells on their way from the DA/hypochord area to the superficial HM. Furthermore, our characterization of the *en1a/en1b*-positive HM fibroblast population indicates that these cells also provide additional proteins that have been shown to be essential for lymphatic development and which do not have an established direct connection to the regulation of Vegfc processing (*cxcl12a* and *svep1*), emphasizing the importance of this cell population as local signaling hubs producing various key regulators of lymphatic migration in the trunk.

The here reported expression of lymphangiogenic factors by fibroblast subpopulations seems to be an evolutionary conserved feature: The publicly available mouse limb muscle dataset of the Tabula Muris single cell transcriptomics database[45], contains an unannotated cell population showing expression of *Adamts3*, *Ccbe1*, and *Vegfc*. Similar to the zebrafish situation, the three genes are only expressed by subpopulations of this cell group. Remarkably, we found that more than 30 orthologs out of the 50 most differentially regulated genes within zebrafish fibroblast cluster 2 (Supplementary Fig. 10) are also expressed in this mouse cell population (including fibroblast-enriched genes such as *Dcn*, *Col1a1*, *Col1a2*, or *Mmp2*[35]). This strongly suggests that a related fibroblast subpopulation to the one we characterize here also exists in muscle tissue of adult mice, and that this cell population can also serve as a source for Vegfc signaling components. Our dataset hence compiles an array of genes and identifies a specific subpopulation of fibroblasts that is not only relevant for zebrafish lymphangiogenesis, but might also be used in other physiological contexts and in other species, including human.

## Methods

**Zebrafish husbandry and strains**. Zebrafish (Danio rerio) strains were maintained according to FELASA recommendations[46]. Animal experiments have been performed according to guidelines of the animal ethics committees at the University of Münster, Germany. Embryonic developmental stages were determined according to reference[47].The following published transgenic lines have been employed in this study: Tg(flt4:mCitrine)[hu7135][48], Tg(flt1[enh]:tdTomato)[hu5333][49], Tg (shh:vegfc-IRES-mTurquoise)[hu10933][10], Tg(kdrl:HRASmCherry)[s916][6] referred to as

*kdrl:mCherry, Tg(mnx1:GFP)[ml2][50], Tg(NBT:dsRed)[zf148][51], Tg(isl1:Gal4FF)[hu6635][52], Tg(UAS:GFP)[nkuasgfp1a], Tg(UAS:RFP)[nkuasrfp1a] [53].*

**Transgenesis.** The *Tg(adamts3:Gal4FF)[mu400]*, *Tg(pdgfra:mCitrine, cmlc2:mTurquoise)[mu401]*, *Tg(vegfc:Gal4FF)[mu402]*, *Tg(vegfc:mCitrine)[mu403]*, *Tg(ccbe1:mCitrine)[hu6741]*, *Tg(ccbe1:Gal4FF)[hu8876]* transgenic reporter lines were generated from BAC DNAs (listed in Supplementary Table 5) following standard BAC recombineering procedures[49]. Briefly, using BAC recombineering in Escherichia coli the long terminal repeats of the Medaka Tol2 transposon were inserted into the BAC clones (in case of the *pdgfra* reporter line containing a cmcl2:mTurquoise expression cassette as a selection marker for transgenesis). In a second step, the respective reporter cassette (mCitrine or Gal4FF) was inserted replacing the first translated ATG of the targeted gene. 150–300 pg of the modified BAC DNA and 25 pg Tol2 mRNA were injected into one-cell embryos.

**Genome editing.** TALEN constructs for the generation of *adamts3[hu10891]*, *adamts3[hu11981]*, *adamts14[hu11304]*, and *adamts2[hu11300]* mutant alleles were assembled following the Golden Gate method and employing the GoldyTALEN modified scaffold[54,55]. The TALEN binding sites in *adamts3* exon 3 were: TAL1, 5′-GCCATGGTGGAATGGCACGAT-3′; TAL2, 5′- AGCTGGAAATGCGACTGATA-3′. The TALEN binding sites in *adamts3* exon 5 were: TAL1, 5′-TGATGATTCTGTCGTCC-3′; TAL2, 5′-AGCATGTGCAGAACT-3′. The TALEN binding sites in *adamts14* exon 3 were: 5′-CTGGGAACAAACATCAACAT-3′; TAL2, 5′-GATCCTGGTGGGATATCGAC-3′. The TALEN binding sites in *adamts2* exon 6 were: TAL1, 5′-TTTTCTCACCAGACAAGA; TAL2, 5′-CATGCAGGGTAATTCATTTT-3′. One-cell embryos were injected with 50–200 pg of the respective mixed TAL1/TAL2 mRNAs. All selected TALEN binding sites flanked a central restriction enzyme site that was used for subsequent screening and efficacy testing.

For the generation of the *adamts2_like[hu11325]* allele CRISPR-mediated genome editing was performed[56]. The sgRNA target site in *adamts2_like* (XM_009303692.3; LOC101883894) exon 2 was 5′-GGGCAATCCTTCTCAGAGTC-3′ and spanned a genomic HinfI restriction enzyme site which was used for subsequent screening and efficacy testing. One-cell stage embryos were injected with 150 pg of zebrafish codon-optimized *nls-zCas9-nls* mRNA and 25 pg *adamts2* sgRNA.

**Genotyping.** *adamts3* and *adamts14* were genotyped by KASPar. *adamts2* and *adamts2_like* were genotyped by PCR and restriction enzyme digestion. *shh:zfvegfc* was genotyped by PCR. The corresponding primers and restriction enzymes are listed in Supplementary Tables 6, 7.

**In situ hybridization (ISH).** Antisense RNA probes for *adamts3* and *adamts14* were generated from cDNA by PCR amplification of a template fragment using primers with a T3 promoter overhang (Supplementary Table 8) followed by in vitro transcription with T3 RNA polymerase using digoxigenin (DIG)-labeled UTP (2 h, at 37 °C). The *vegfc* probe was generated from a plasmid template[57] using T7 RNA polymerase. ISH was carried out following standard procedures[58] with the following changes: After fixation (4% paraformaldehyde in PBS, overnight at 4 °C) and storage in 100% methanol, the 26/36/48 hpf embryos were treated with proteinase K for 10 min at room temperature at a concentration of 10 μg/mL. After re-fixation (4% paraformaldehyde, 20 min), washing with PBST and pre-hybridization at 65 °C, embryos were incubated in hybridization buffer containing DIG-labeled antisense probes at 65 °C for at least 12 h. For the detection of *vegfc*, 100 ng of the column-purified probe were added to the hybridization buffer. For *adamts3* and *adamts14* transcript detection, 100 ng of each of the respective probes 1–3 (Supplementary Table 8) were combined and used for the hybridization. All subsequent washing steps, the incubation with anti-DIG primary antibody (sheep anti-Digoxigenin-AP Fab Fragments, 1:2000, Roche, cat-no. 11093274910, Lot: 16646822) and the alkaline-phosphatase-conjugated secondary antibody as well as the staining reaction using NBT/BCIP substrates was performed according to the published protocol. Embryos were imaged on a Nikon Eclipse Ni microscope.

**RNAscope.** RNAscope probes for *adamts3*, *adamts14*, *ccbe1*, and *vegfc* (Supplementary Table 9) were designed and synthesized by ACD, a bio techne brand. The RNAscope transcript detection was performed on embryos that were fixed in 4% paraformaldehyde at RT for 20 min using the RNAscope Fluorescent Multiplex Reagent Kit (ACD) and following standard procedures[26]: Fixed embryos were washed three times for 5 min in 1 ml PBT (0.1% Tween), followed by a series of dehydration steps: 25, 50, 75, and 100% methanol for 5 min each. After overnight incubation in 100% methanol at −20 °C the solution was completely removed and the embryos were air-dried at room temperature for 30 min. Two drops of Pretreat 3 were added and the embryos were incubated at room temperature for 20 min. The individual probes were simultaneously heated up to 40 °C in the water bath for 10 min and were brought back to room temperature before the respective C1, C2, and C3 probes were mixed in a 50:1:1 ratio with a final volume of 50 μl per tube. After Pretreat 3 incubation, embryos were washed three times with PBT at room temperature followed by an incubation in 50 μl of mixed target probes at 40 °C overnight. After three washes with 0.2X SSCT for 15 min each at room

temperature, embryos were re-fixed in 4% PFA for 10 min at room temperature and washed three times 15 min with 0.2x SSCT at room temperature. Embryos were subsequently incubated in two drops of Amp1 solution from the kit for 30 min at 40 °C followed by three 15 min washing steps in 1 ml 0.2X SSCT at room temperature. Analogous incubation and washing steps were performed for Amp2 (15 min incubation), Amp3 (30 min incubation), and Amp4 (15 min incubation) solution. After the final wash with 0.2X SSCT at room temperature, embryos were stained in two drops of DAPI solution from the kit at 4 °C overnight, washed with PBT and imaged on a Leica SP8 confocal microscope.

**Immunohistochemistry.** Antibody staining was performed according to a previously published protocol[59]. In brief, embryos were fixed in 4% PFA overnight at 4 °C, washed three times 5 min with 100% methanol on ice and incubated in methanol with 3% $H_2O_2$ for 60 min on ice. Afterwards, embryos were washed three times 5 min with 100% methanol on ice and then kept in 100% methanol at −20 °C for at least 2 days before use. Embryos were washed with PBT (0.1% Tween 20) three times for 10 min and incubated in PBT with 30% sucrose (w/v) overnight at 4 °C. After three more washes with PBT for 10 min each, embryos were incubated in 150 mM Tris-HCl (pH 9.0) for 5 min, followed by an incubation at 70 °C for 15 min. The embryos were washed two times with PBT 10 min and with distilled water twice for 5 min each. Afterwards, embryos were transferred to ice-cold acetone and incubated at −20 °C for 40 min. Then, embryos were washed two times with PBT 5 min before they were treated with 10 μg/ml Proteinase K in PBT for 10 min at room temperature and re-fixed with 4% PFA at room temperature for 20 min. After three more washing steps in PBT for 5 min each embryos were rinsed with cold TBST (0.1% Triton X-100), followed by a blocking step with 1% BSA and 10% goat serum in TBST overnight at 4 °C. Primary antibodies in blocking buffer were applied to the embryos, which were then incubated overnight at 4 °C on a shaker. Embryos were washed five times with TBST for 30 min each, once with Maleic buffer (150 mM Maleic acid; 100 mM NaCl; 0.001% Tween 20 pH 7.4) for 30 min at room temperature, followed by an incubation in 2% blocking reagent (Roche) in Maleic buffer at RT for 2–3 h. The secondary antibody in 2% blocking reagent was added and the embryos were incubated overnight at 4 °C. On the following day, embryos were washed with Maleic buffer five times 30 min each and rinsed with PBS. After the washing steps, PBS was removed and 100–150 μl tyramide signal amplification reagent was added. The embryos were incubated at room temperature for 3 h in dark before they were washed with TBST several times during the next 1–2 days at room temperature on a shaker before imaging and analysis.

The following antibodies were used at the indicated dilutions: mouse anti-engrailed 4D9 primary (1:4; Developmental Studies Hybridoma Bank, University of Iowa, Antibody Registry ID: AB_528224) with Peroxidase AffiniPure Goat Anti-Mouse IgG, light chain specific secondary (1:100; Jackson Immuno Research, cat-no. 715-035-150) followed by Alexa568-tyramide signal amplification reagent (TSA kit, PerkinElmer, cat-no. 2305851); chicken anti-GFP primary (1:400, Abcam, cat-no. ab13970) with Alexa488-conjugated goat anti-chicken secondary (1:200, Invitrogen, cat-no. A11039).

**Cell transplantation assay.** Heterochronic cell transplantations were performed by using 4 hpf donor and 6 hpf host embryos. Approximately 20 cells were taken from the donor's animal pole and transferred close to the host's lateral marginal zone (for fibroblast cells), or slightly above for neuronal cells. Donor embryos were injected with dextran-Alexa647 to identify the transplanted cells in host embryos. In addition, *adamts3* wild-type donor embryos positive for Tg(adamts3:gal4;UAS:GFP) were used to highlight the *adamts3*-expressing cells within the transplant.

**Protein purification and VEGFC-processing assay.** For purification of ADAMTS3 and ADAMTS14 recombinant proteins, a previously published protocol was adopted[60]. Stably transfected cells were grown in DMEM supplemented with 10% fetal calf serum and hygromycin. Upon confluence, the growth medium was replaced by serum free DMEM containing soybean trypsin inhibitor (40 μg/ml), heparin (5 μg/ml), and $ZnCl_2$ (80 μM). After 48 h the medium was collected, concentrated, and subsequently loaded on a 25 ml Concanavalin A-Sepharose column that was pre-equilibrated in buffer A (50 mM Tris pH 7.5; 1 M NaCl; 2 mM $CaCl_2$). After a washing step with buffer A, protein elution was performed using buffer A supplemented with 500 mM α-methyl-D-mannoside. The fractions containing recombinant enzymes were then dialyzed against buffer B (50 mM Tris pH 7.5; 0.2 M Na $Cl_2$) before they were loaded on a 5 ml heparin-Sepharose column, washed with buffer B and finally eluted in buffer A. Note that the precise concentration of active ADAMTS proteins cannot be determined accurately since each enzyme preparation consists of several co-purifying polypeptides with different molecular weights and different activities. Quantifications of purified proteins were made after SDS-PAGE and staining with SYPRO Orange using known concentrations of BSA as reference. Only the product around 105 kDa was taken into account, as this polypeptide corresponds to the expected molecular weight of the fully mature ADAMTS form. Values of 8 and 12 ng/μl were obtained for ADAMTS3 and ADAMTS14, respectively.

HEK293 cells expressing human pro-VEGFC were cultured for 48 h in DMEM containing 0.1% FBS, 0.2 mM AEBSF (serine protease inhibitor), and 50 μM

Decanoyl-RVKR-CMK (furin inhibitor). For the processing assay, 20 μl of conditioned medium were supplemented with NaCl (0.3 M final concentration) and 8 μl of either ADAMTS3, ADAMTS14 or control buffer in the absence or presence of 25 mM EDTA (used as an inhibitor of metalloproteinases) in a total volume of 40 μl. After an 18 h incubation at 37 °C, samples were denatured in reducing conditions (1x Laemmli sample buffer containing 200 mM DTT). Western blotting analysis was performed using a polyclonal goat anti-human VEGFC primary antibody (1:200, R&D, AF752). Note that the VEGFC antibody does not react with the 29kDa C-terminal domain of VEGFC and shows relatively low affinity for the fully processed mature 21 kDa form.

**FACS**. Approximately 100 trunks of the respective transgenic 48 hpf embryos were incubated in 600 μl autoclaved PBS at 30 °C for 15 min. After a centrifugation at 600 rpm for 4 min the supernatant was discarded and the trunk was subsequently partially dissociated in 1 ml 0.5% trypsin-EDTA (Gibco) supplemented with 10 U DNaseI (Promega) for 30 min at 30 °C. During this incubation step, the solution was pipetted up and down three times every 10 min. After addition of 300 μl PBS (to dilute the trypsin) the mixture was centrifuged at 900 rpm at room temperature for 5 min. The pellet was resuspended in 300 μl PBS and filtered through a Falcon 40 μm cell strainer to exclude undissociated tissue fragments using a gentle centrifuge step at 600 rpm for 30 s. Under these conditions the trunks were not completely dissociated thereby enriching for cells that were located closer to the surface while decreasing the proportion of cells in the solution that originated from more central regions within the trunk. Dissociated cells were sorted directly into lysis buffer using a FACSAria IIIu cell sorter (BD Biosciences) with FACSDiva 8.0.1 (https://www.bdbiosciences.com/) and FlowJo 10.6.1 (https://www.flowjo.com/) software. For pooled cell populations, RNA extraction and sequencing were carried out by the Core Facility Genomics Münster.

**Smart-seq2 library preparation and sequencing**. Single-cell cDNA libraries were prepared as described before[61]. In brief, cDNA was transcribed from mRNA using oligo(dT) primer and SuperScript II reverse transcriptase (ThermoFischer). Second strand cDNA was synthetized using a template switching oligo. The double stranded cDNA was then amplified by PCR for 24 cycles. The quality and quantity of purified cDNA was controlled using a 2100 BioAnalyzer with a DNA High Sensitivity Chip (Agilent Biotechnologies) and Qubit (ThermoFischer). Upon passing quality control (QC), the cDNA was fragmented using Tn5 transposase, and each single well was uniquely indexed using the Illumina Nextera XT index kits (Set A–D). The indexed cDNA libraries from one 384-well plate were pooled into one sample and sequenced on one lane of a HiSeq3000 sequencer (Illumina), using dual indexing and single 50 base-pair reads.

**Sequence analysis and data processing**. After sequencing of pooled cDNA libraries as described above, the sequences were de-multiplexed into single-cell fastq files applying standard parameters of the Illumina pipeline (bcl2fastq) using Nextera index adapters. The individual fastq files were mapped to the zebrafish reference genome GRCz11, using TopHat2 with Bowtie2 option[62,63] where adapter sequences were removed using trim galore before read mapping. Doublets were removed using the samtools software. The generated BAM files containing the results of the alignment were sorted according to the mapping position, and raw read counts for each gene were calculated using featureCounts function from the Subread software package[64]. As technical control, 92 artificial RNA sequences (ERCCs) were added to the lysis buffer and in the mapping pipeline.

After mapping and gene counting, the data for the individual cells was combined into an expression matrix dataset showing raw counts per gene of each individual cell as input data. The annotation of the ENSEMBL identifiers was done using the org.Dr.eg.db (version 3.7.0) package in R-software, retaining the ERCC counts to be used as technical controls in the dataset.

The resulting expression data were analyzed using the software packages velocyto (http://velocyto.org)[65], SingleCellExperiment[66] and pathway and gene set overdispersion analysis (pagoda2) (https://github.com/hms-dbmi/pagoda2)[67], applying standard QC, normalization and analysis procedures. In brief, the expression matrix was structured and organized using the SingleCellExperiment R-software package. Low quality cells were filtered in a stepwise manner. First, cells with low library sized (<50,000 counts), and number of expressed genes (≤1500) were removed from the dataset. If necessary, also cells with high percentage of reads mapped to ERCCs (>10 %) were removed. Cells with an unusual high number of expressed genes (>10,000) were removed as potential doublets. Non- or low-expressed genes were removed, due to the criteria: gene expression in at least three cells with a cumulative counts value of 300, considering the complete dataset. After QC, the cells were normalized to 500,000 counts total library size. For clustering, the QC- and gene expression-filtered raw counts expression matrix was loaded into the pagoda2 R-software package. PCA analysis was performed in pagoda2 using the attributes (nPca = 50, n.odgenes = 3000), followed by nearest neighbor clustering. For visualization t-stochastic neighbor embedding (t-SNE) was used as embedded functions in the pagoda2 or SingleCellExperiment R-software package. For cell-differentiation prediction (RNA-velocity), the velocyto software package was applied using standard parameters. For the co-expression analysis, a threshold of ≥20 counts per cell was used for adamts3, adamts14, ccbe1, and vegfc to qualify as an active expressing cell.

**Microscopy**. Confocal imaging was performed on living embryos embedded in 0.8% low melting point agarose (Invitrogen) with 0.0168% tricaine (Sigma A-5040) on a Leica SP8 microscope employing Leica LAS X 3.5.6.21594 software (https://www.leica-microsystems.com/). Images were processed using Adobe Photoshop CC 2017 (https://www.adobe.com/products/photoshop.html) and Fiji [(Fiji Is Just) ImageJ2.0.0-rc-43/1.52 u](http://fiji.sc/Fiji). Stitching of composite pictures was performed using Adobe Illustrator CC 2015 (https://www.adobe.com/products/illustrator.html). 3D-reconstructions of z-stacks were done with Bitplane Imaris 8.2.1 (https://imaris.oxinst.com/). For light microscopy, a Nikon Eclipse Ni microscope with NIS-Elements 4.40.00 software (https://www.microscope.healthcare.nikon.com/) was employed.

**Statistics and reproducibility**. The following section indicates how often experiments have been repeated independently with similar results. The phenotypic analysis of adamts3 and adamts14 single and double mutants (Fig. 1c–e, g–i, l–q, Supplementary Fig. 1d, e) was performed at least five times, quantifications of PLs and vISVs (Fig. 1k) were done at least two times and the analysis of all other single or double mutant combinations (Fig. 1j and Supplementary Fig. 1b, c) was done at least two times. The in vivo zebrafish Vegfc-processing assays (Fig. 2b–i) were done at least three times each while the in vitro processing assay (Fig. 2j) was performed five times. The adamts3 ISH (Fig. 3a–c) was done three times and the adamts3: Gal4FF transgene analysis was performed five times (Fig. 3d, i–m) and two times in conjunction with the motoneuronal marker mnx1:GFP (Fig. 3e, f). adamts14 expression analysis by ISH and RNAscope (Fig. 4) was performed at least three times each. Cell transplantation assays for adamts3 (Fig. 5) and adamts14 (Fig. 6) were conducted at least 13 times each. Transgene co-expression (Fig. 7c–p) was assessed three independent times and the expression pattern of the pdgfra reporter alone and in combination with other transgenes (Fig. 8) was analyzed at least three times. Rescue experiments with a mature VEGFC construct (Fig. 10) were performed six times. The axonal pattern of motoneurons in adamts3;adamts14 double mutants (Supplementary Fig. 2) was analyzed at least two times for the indicated time points. Expression analysis for vegfc by ISH (Supplementary Fig. 4) was performed once and the expression analysis for vegfc, ccbe1, adamts14 and pdgfra by RNAscope (Supplementary Figs. 5–7) was done at least two times. The Engrailed antibody stainings have been performed twice for all combinations (Supplementary Fig. 11a–i) and the ISH against nuak1b (Supplementary Fig. 11k, l) and itgbl1 (Supplementary Fig. 11n, o) were done two times.

All images shown in the figures are representative examples of the respective phenotypes and expression patterns. Statistical analysis of PL and vISV quantifications was performed with GraphPad Prism 8 (https://www.graphpad.com/scientific-software/prism/).

**Reporting summary**. Further information on research design is available in the Nature Research Reporting Summary linked to this article.

## Data availability

The authors declare that the data supporting the findings of this study are available within the paper and its supplementary information files. The raw single-cell RNA sequencing data of this study have been deposited in the NCBI's Gene Expression Omnibus database under the accession code: GSE146923.

Source data underlying Figs. 1k, 2j, and 7b are provided as a Source Data file.

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

## Acknowledgments

This work was supported by FOR2325 (G.W., A.v.I., S.S.-M., F.l.N.) and CRC1348 (DFG, project B08, Y.P. and S.S.-M.). We thank R. Adams for support with FACS analysis, T. Zobel (Imaging Network Münster) for help with the RNAscope image analysis and B. Ponsioen for initial characterization of the *ccbe1* transgenic line. K. Weiz generated BAC constructs used for transgenesis. We thank the Single Cell Sequencing Core lab (ICMC, KI—Huddinge) for their help with library preparation and sequencing and the Raz lab for their help with RNAscope.

## Author contributions

G.W. and A.v.I. performed experiments, analyzed the data, created the figures, and wrote the paper. L.M. and C.B. performed, analyzed, and discussed the scRNAseq experiments. J.P.-M. generated the *ccbe1* reporter transgenes and Y.P. established the *pdgfra* reporter line. L.D. and A.C. performed the VEGFC-processing assay. FAC-sorting was conducted by M.S., F.l.N. provided constructs and fish lines and discussed the conceptual framework. S.S.-M. acquired funding, reviewed, and edited the paper. S.S.-M. and A.v.I. conceived and designed experiments and supervised the project. All authors provided critical comments and reviewed the paper.

## Competing interests

The authors declare no competing interests.
