## [Peer Review File · Nature Communications]

Reviewers' Comments:

Reviewer #1:

Remarks to the Author:

Summary:

In their manuscript, Wang and colleagues investigate the processing of the lymphangiogenic growth factor Vegfc in the zebrafish embryo, focussing on Adamts proteases. The authors report that Adamts3 and its prologue Adamts14 function redundantly to process Vegfc into its active form. As a result, lymphatic structures (in the zebrafish trunk) fail to form when both genes are inactivated. They further identify specific neuronal cells and fibroblast (sub-)populations as sources of these secreted proteases, which also express other components of the Vegfc-processing machinery. Based on all these data, the authors propose that Vegfc processing is spatially restricted and that this restriction is critical to induce coordinated lymphatic sprouting.

Comment:

This is an elegant study that provides detailed insights into the biology of the essential lymphangiogenic growth factor Vegfc. Specifically, the authors identify crucial components of the Vegfc-processing machinery, its expression and function in the developing zebrafish embryo.

The results are not necessarily conceptually novel but still significant and insightful. Importantly, the manuscript is carefully written, and the experiments are of the highest quality, using state-of-the-art tools, reagents and technologies.

The only aspect the authors might want to change is the length of the paper. In its present form, some paragraphs are rather long, which dilutes some of the take-home messages. This refers particularly to the description of the expression analysis. I feel that the manuscript would benefit if some of these parts are shortened.

Nevertheless, I am enthusiastic about this work and its appeal to the vascular biology community.

Reviewer #2:

Remarks to the Author:

The manuscript by Wang et al describes the redundant function of two procollagen N-proteinases, Adamts14 and Adamts3 in lymphatic development, in particular, their role in processing VEGF-C ligand. Previously, Schulte-Merker group has identified key factors of lymphatic development in zebrafish, most notably ccbe1. This work continues their efforts to further delineate the molecular basis of lymphatic development using zebrafish as a model system. The authors showed that Adamts3 and Adamts14 redundantly modulates the availability of VEGF-C, thereby regulating lymphatic development in zebrafish. While Adamts3 has been shown to be critical regulator for VEGF-C processing in mice by Mark Kahn's group, the role of Adamts14 has not been fully elucidated. The authors then delineated the source of Adamts3 and Adamts14 with elegantly designed cell transplantation experiments, and single cell analysis. While the findings of the manuscript that Adamts14 and Adamts3 are essential component for VEGF-C processing is important, a number of issues should be resolved prior to the publication.

1. While the phenotype of zebrafish is convincingly and elegantly demonstrated, there is no indication that the authors actually tested Adamts14 can process VEGF-C. Although the authors elegantly demonstrated the importance of Adamts14 in lymphatic development, it is formally possible that the effects of Adamts14 could be purely indirect but not on VEGF-C itself. Biochemical validation would be necessary.

2. The majority of the data presented here is focused on regulation of gene transcription. Considering that both Adamts3 and Adamts14 are secreted proteins, it is equally important to show the distribution of mature proteins.

3. The image quality in Fig.1 and Fig.2 could be improved. Could the authors use better arterial markers other than Flt1:mCit? Alternatively, diagram of the structure would help the audience to follow the figures.
4. The coexpression of markers were only examined by transgenes. Considering the half-life of fluorescent protein, it is desirable to directly show co-expression by double in situ or antibody staining.
5. The authors isolated vegfc:Gal4FF;UAS:GFP positive cell by FACS and used for single cell analysis. Surprisingly, the single cell data revealed that only a fraction of isolated cells are positive for vegf-c. It is understandable that Gal4-UAS system provides distorted information on the expression of target gene, it is quite surprising and disturbing that vegf-c expression is not universally present in the isolated population. This finding could significantly undermine the validity of single cell analysis presented in the manuscript.
6. Also, it is quite difficult to understand 'partial dissociation' described in the manuscript for single cell preparation.
7. Single cell analysis is costly and time-consuming, yet curiously, the authors did not go in to details on the subpopulation they identified and only included cursorily-performed analyses in the manuscript. The detailed analysis and validation on this interesting subpopulation is needed.

We thank both reviewers for their time and effort. Please find below our point by point response.

Reviewer #1 (Remarks to the Author):

Summary:

In their manuscript, Wang and colleagues investigate the processing of the lymphangiogenic growth factor Vegfc in the zebrafish embryo, focussing on Adamts proteases. The authors report that Adamts3 and its prologue Adamts14 function redundantly to process Vegfc into its active form. As a result, lymphatic structures (in the zebrafish trunk) fail to form when both genes are inactivated. They further identify specific neuronal cells and fibroblast (sub-)populations as sources of these secreted proteases, which also express other components of the Vegfc-processing machinery. Based on all these data, the authors propose that Vegfc processing is spatially restricted and that this restriction is critical to induce coordinated lymphatic sprouting.

Comment:

This is an elegant study that provides detailed insights into the biology of the essential lymphangiogenic growth factor Vegfc. Specifically, the authors identify crucial components of the Vgfc-processing machinery, its expression and function in the developing zebrafish embryo.

The results are not necessarily conceptually novel but still significant and insightful. Importantly, the manuscript is carefully written, and the experiments are of the highest quality, using state-of-the-art tools, reagents and technologies.

The only aspect the authors might want to change is the length of the paper. In its present form, some paragraphs are rather long, which dilutes some of the take-home messages. This refers particularly to the description of the expression analysis. I feel that the manuscript would benefit if some of these parts are shortened.

We thank the reviewer for the positive assessment. We appreciate that the manuscript length is above average, and have made a serious effort to shorten the manuscript. We have also discussed this issue with the editor.

Nevertheless, I am enthusiastic about this work and its appeal to the vascular biology community.

--

Reviewer #2 (Remarks to the Author):

The manuscript by Wang et al describes the redundant function of two procollagen N-proteinases, Adamts14 and Adamts3 in lymphatic development, in particular, their role in processing VEGF-C ligand. Previously, Schulte-Merker group has identified key factors of lymphatic development in zebrafish, most notably ccbe1. This work continues their efforts to further delineate the molecular basis of lymphatic development using zebrafish as a model system. The authors showed that Adamts3 and Adamts14 redundantly modulates the availability of VEGF-C, thereby regulating lymphatic development in zebrafish. While Adamts3 has been shown to be critical regulator for VEGF-C processing in mice by Mark Kahn's group, the role of Adamts14 has not been fully elucidated. The authors then delineated the source of Adamts3 and Adamts14 with elegantly designed cell transplantation experiments, and single cell analysis. While the findings of the manuscript

that Adamts14 and Adamts3 are essential components for VEGF-C processing is important, a number of issues should be resolved prior to the publication.

We thank the reviewer for the critical assessment, and have incorporated, wherever possible, the suggestions, as delineated below.

1. While the phenotype of zebrafish is convincingly and elegantly demonstrated, there is no indication that the authors actually tested Adamts14 can process VEGF-C. Although the authors elegantly demonstrated the importance of Adamts14 in lymphatic development, it is formally possible that the effects of Adamts14 could be purely indirect but not on VEGF-C itself. Biochemical validation would be necessary.

We have, while the manuscript was under review, performed in vitro VEGFC processing assays using partially purified human proteins and human pro-VEGF-C. We show now, that besides the known processing activity of ADAMTS3, also ADAMTS14 can proteolytically process VEGF-C in conditioned medium from HEK-293T cells. These data are incorporated in Figure 2, where we also show that in an ectopic over-expression assay, zebrafish Adamts14 can process zebrafish as well as human VEGF-C in vivo. This, together with the now provided in vitro data, makes it extremely likely that Adamts14, like Adamts 3, acts as a processing enzyme.

2. The majority of the data presented here is focused on regulation of gene transcription. Considering that both Adamts3 and Adamts14 are secreted proteins, it is equally important to show the distribution of mature proteins.

We agree with the reviewer that following the secreted proteins in the extracellular space would be interesting and possibly insightful, not only for Adamts proteases but also for Vegfc and Ccbe1 proteins. However, antibody stainings in zebrafish are extremely challenging in general since the vast majority of available antibodies do either not cross-react with the respective zebrafish proteins, or do not specifically stain the respective protein. For zebrafish Adamts3 and Adamts14 we are not aware of any commercially available antibody that could detect the protein in situ, neither for the murine nor the zebrafish version. The situation is further complicated, since the maturation process of Adamts3 (and most likely also Adamts14) involves a N- and C-terminal cleavage of the protein itself (Brouillard et al., 2017). Hence, only antibodies that are able to discriminate between the unprocessed and the mature form of the proteins would be fully informative. Such antibodies are not available.

We would like to stress here that even for the well-studied Vegfc protein the actual tissue distribution of the mature protein is still elusive in all animal models, for the same reasons. Of note, the transgenic zebrafish line we present here for the first time might present the hitherto best tool to study *vegfc* mRNA expression in a vertebrate model.

3. The image quality in Fig.1 and Fig.2 could be improved. Could the authors use better arterial markers other than Flt1:mCit? Alternatively, diagram of the structure would help the audience to follow the figures.

Thank you for pointing this out. We have now improved both figure 1 and 2 by replacing several pictures with new ones. We also added additional labels for vascular structures that can be seen in the individual pictures, which should make it easier for the reader to follow the figures.

The widely used *flt1:tdTomato* transgene represents an artificial combination of enhancer elements of the *flt1* promoter, that have been selected to achieve the most specific

expression of tdTom in arterial intersegmental vessels (Bussmann et al., 2010). Although expression levels within the DA are rather low, this transgene facilitates the discrimination between arterial and venous intersegmental vessels which is instrumental for the analysis of mutants affecting secondary sprouting.

4. The coexpression of markers were only examined by transgenes. Considering the half-life of fluorescent protein, it is desirable to directly show co-expression by double in situ or antibody staining.

We have performed double and triple fluorescent in situs using the RNAscope technique and show co-expression for several transcript combinations in the new supplementary figure 4. While the results are fully consistent with the transgenic analysis, we experienced a limitation caused by the very tight packing of cells and the irregular shape of the fibroblasts at the horizontal myoseptum. While it was still possible to show qualitatively the co-expression of genes (thereby confirming our co-expression analysis based on transgenic reporter lines), the close juxtaposition of nuclei did not allow a quantitative statement about the degree of overlap between the expression pattern of the individual players. In fact, this limitation of the technique was the reason for us to turn to the scRNAseq analysis, which allowed us to tackle this question by assessing directly the numbers of cells within a specific subpopulation/cluster that are positive for a given combination of transcripts. The transcript analysis also re-confirmed the co-expression of marker genes.

5. The authors isolated *vegfc:Gal4FF;UAS:GFP* positive cell by FACS and used for single cell analysis. Surprisingly, the single cell data revealed that only a fraction of isolated cells are positive for *vegfc*. It is understandable that Gal4-UAS system provides distorted information on the expression of target gene, it is quite surprising and disturbing that *vegfc* expression is not universally present in the isolated population. This finding could significantly undermine the validity of single cell analysis presented in the manuscript.

There are several points we want to bring up here.

- (1) First of all, we apologize for not having stated clearly enough in the original version of the manuscript that not all cells that were used for the scRNAseq analysis originated from a *vegfc* reporter sorting event. In fact we also used a considerable amount of cells sorted positive for *pdgfr α* (to isolate a broader fibroblast population) and a control endothelial population that was sorted for *kdrl:mCherry* expression (while being negative for the *vegfc* reporter). Hence, one cannot per se expect to see *vegfc* transcripts in all cells that are depicted in the scRNAseq plots. We tried to make this more obvious now for the reader in the revised version of the manuscript.
- (2) Another important fact that one needs to consider is the inherent nature of single-cell transcriptome data. Whereas the scRNAseq method of our choice (SmartSeq2) is known to provide the deepest capture of the single cells transcriptome of all methods "on the market" (about 30% of the cells' mRNA molecules are captured on average, which is about twice as high compared to commercial droplet-based method (10x)), the lack of reads for any particular transcript in an individual cell does not mean that that cell did not contain the transcript - it just failed to be captured in the stochastic cDNA synthesis process. However, if the cell negative for e.g. *vegfc* forms part of a homogeneous cluster (such as those that we identify using *pagoda2*) in which a substantial proportion of the other cells' transcriptomes provided sequence reads for

vegfc, the assumption would be that all cells of that cluster express *vegfc*. Of course, putative burstiness of the transcription may mean that not all cells of a certain cluster (or type) contain the transcript in question at any given moment in time, providing another reason for cell-to-cell variation in capture rate for individual transcripts, in spite of high overall transcriptional similarity.

- (3) Related to point (2), and as the reviewer correctly points out, transcript and the actual protein levels for a given gene might not always match up precisely in a temporal manner, especially if a gene is dynamically regulated while the protein is relatively stable. The use of transgenic reporters can enhance this discrepancy due to the protein stability of – in our case - the Gal4FF and the GFP protein, hence making it difficult to appreciate dynamic changes in gene expression levels in real time. In consequence, some of the FAC sorted *vegfc*:Gal4FF, UAS:GFP positive cells might actually not actively express *vegfc* transcripts anymore and would therefore be negative in the *vegfc* transcript plot.

To summarize points (1-3): the major advantage of scRNAseq (compared to RNAseq on pooled cell populations, for example) is that all subsequent analysis and its interpretation are based on the actual transcriptomes of the individual cells, and are therefore independent of the reporter transgene expression. In other words, even if the cell population after FACS were not homogeneous, this would not impact on the scRNAseq analysis.

As a final clarification about our statements concerning co-expression in the single-cell data: For these we only count the cells that gave sequence reads for e.g. *vegfc* transcripts, and we then checked how many of those cells also provided sequence reads for *ccbe1*. Hence, all quantitative statements about co-expression (e.g. 80% of *vegfc*+ cells express also *ccbe1*) are conservative, i.e. the real co-expression frequency is most likely higher.

6. Also, it is quite difficult to understand ‘partial dissociation’ described in the manuscript for single cell preparation.

Thank you for pointing that out. We have added a section in the Material and Methods part that describes in more detail how the cell suspension for FACS and subsequent scRNAseq analysis has been prepared. We have also reworded the description in the Result section.

7. Single cell analysis is costly and time-consuming, yet curiously, the authors did not go in to details on the subpopulation they identified and only included cursorily-performed analyses in the manuscript. The detailed analysis and validation on this interesting subpopulation is needed.

We have now expanded the scRNAseq analysis by providing additional information about the fibroblast subpopulation at the horizontal myospetum. Besides providing a heatmap that indicates the relative expression of transcripts most specific for each pagoda2 cell cluster, we added three list of the most differentially regulated genes in cluster 2 when compared to 1) all other isolated cells, 2) all other fibroblast clusters or 3) the most closely related progenitor cluster1. Using this extra source of information on the fibroblast subpopulation in question, we furthermore show *in situ* data for two example genes of this list which validates (together with the Engrailed antibody stainings) that the cells in cluster 2 represent the fibroblast subpopulation localized at the HM region.

Reviewers' Comments:

Reviewer #2:

Remarks to the Author:

The authors satisfactorily addressed all of my previous concerns. The revised manuscript was much improved now. In particular, the key figures are clearly shown and the data analysis was done in depth.